# A microbial supply chain for production of the anti-cancer drug vinblastine

Jie Zhang[1], Lea G. Hansen[1], Olga Gudich[1], Konrad Viehrig[1], Lærke M. M. Lassen[1], Lars Schrübbers[1], Khem B. Adhikari[1], Paulina Rubaszka[1], Elena Carrasquer-Alvarez[1], Ling Chen[1], Vasil D'Ambrosio[1], Beata Lehka[1], Ahmad K. Haidar[1], Saranya Nallapareddy[1], Konstantina Giannakou[1], Marcos Laloux[1], Dushica Arsovska[1], Marcus A. K. Jørgensen[1], Leanne Jade G. Chan[2,3], Mette Kristensen[1], Hanne B. Christensen[1], Suresh Sudarsan[1], Emily A. Stander[4], Edward Baidoo[2,3], Christopher J. Petzold[2,3], Tune Wulff[1], Sarah E. O'Connor[5], Vincent Courdavault[4], Michael K. Jensen[1✉] & Jay D. Keasling[1,2,3,6,7✉]

Monoterpene indole alkaloids (MIAs) are a diverse family of complex plant secondary metabolites with many medicinal properties, including the essential anti-cancer therapeutics vinblastine and vincristine[1]. As MIAs are difficult to chemically synthesize, the world's supply chain for vinblastine relies on low-yielding extraction and purification of the precursors vindoline and catharanthine from the plant *Catharanthus roseus*, which is then followed by simple in vitro chemical coupling and reduction to form vinblastine at an industrial scale[2,3]. Here, we demonstrate the de novo microbial biosynthesis of vindoline and catharanthine using a highly engineered yeast, and in vitro chemical coupling to vinblastine. The study showcases a very long biosynthetic pathway refactored into a microbial cell factory, including 30 enzymatic steps beyond the yeast native metabolites geranyl pyrophosphate and tryptophan to catharanthine and vindoline. In total, 56 genetic edits were performed, including expression of 34 heterologous genes from plants, as well as deletions, knock-downs and overexpression of ten yeast genes to improve precursor supplies towards de novo production of catharanthine and vindoline, from which semisynthesis to vinblastine occurs. As the vinblastine pathway is one of the longest MIA biosynthetic pathways, this study positions yeast as a scalable platform to produce more than 3,000 natural MIAs and a virtually infinite number of new-to-nature analogues.

Plants produce some of the most potent human therapeutics and have been used for millennia to treat illnesses. The monoterpene indole alkaloids (MIAs) are plant secondary metabolites mostly found in Gentianales plants, bolstering a notable structural diversity and pharmaceutically valuable biological activities, with more than 3,000 MIAs derived from their common precursor strictosidine[4,5]. The MIA chemotherapeutic drugs irinotecan, vinblastine and vincristine are included in the World Health Organization list of essential medicines and exemplify some of the few established MIA therapeutics on the market, with annual market sizes of more than €1 billion (ref. [6]). Despite the vast repertoire of MIAs with documented bioactivities, obtaining requisite amounts of MIAs needed for further development and clinical applications is often challenging because of the low yields observed in extracts from natural plant resources, as exemplified by the chemotherapeutic drugs vinblastine and vincristine requiring 500 and 2,000 kg of *Catharanthus roseus* dried leaves, respectively, to obtain 1 g of products[3,7,8]. Likewise, owing to the numerous stereo centres found in MIAs, multi-step synthetic chemistry holds little promise for bulk production

of complex MIA therapeutics[2,9], and thus the current vinblastine supply chains rely on extraction from *C. roseus* and subsequent condensation of the catharanthine and vindoline precursors[10]. As vincristine and vinblastine were listed as being drugs with a shortage in 2019–2020 by the US Food and Drug Administration[11], there is a growing awareness of the importance of refactoring the biosynthesis of MIAs in genetically tractable heterologous hosts.

Refactoring complex plant biosynthetic pathways of human therapeutics in microbial hosts could promise more access to plant natural products and new-to-nature derivatives thereof through scalable fermentation[12]. Indeed, baker's yeast *Saccharomyces cerevisiae* has been engineered to produce plant-derived opiates, cannabinoids, terpenes and alkaloids to treat cancer, pain, malaria and Parkinson's disease[12–16]. Whereas small parts of the vinblastine biosynthetic pathway have been refactored into yeast[4,17–19], only recently was the full 31-step vinblastine biosynthetic pathway from *C. roseus* explained[20,21], opening up the possibility for a demonstration of a microbial supply chain for vinblastine using engineered yeast.

[1]Novo Nordisk Foundation Center for Biosustainability, Technical University of Denmark, Kongens Lyngby, Denmark. [2]Joint BioEnergy Institute, Emeryville, CA, USA. [3]Biological Systems and Engineering Division, Lawrence Berkeley National Laboratory, Berkeley, CA, USA. [4]EA2106 Biomolecules and Plant Biotechnology, University of Tours, Tours, France. [5]Department of Natural Product Biosynthesis, Max Planck Institute for Chemical Ecology, Jena, Germany. [6]Department of Chemical and Biomolecular Engineering, Department of Bioengineering, University of California, Berkeley, CA, USA. [7]Center for Synthetic Biochemistry, Institute for Synthetic Biology, Shenzhen Institutes of Advanced Technologies, Shenzhen, China. ✉e-mail: mije@biosustain.dtu.dk; keasling@berkeley.edu

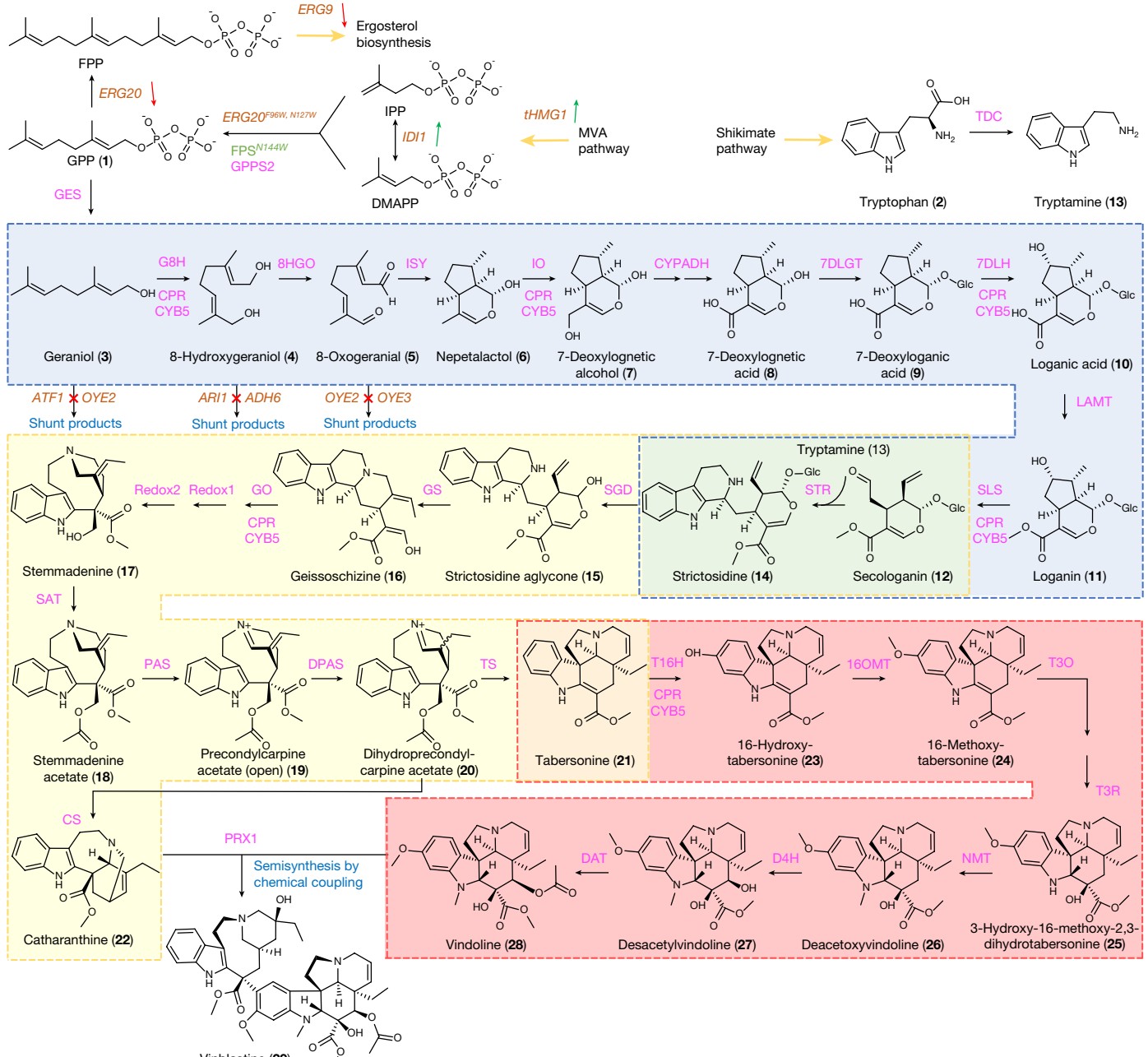

**Fig. 1 | Complete biosynthetic pathway for the production of vinblastine in yeast.** Yeast genes (orange) overexpressed, dynamically knocked down or deleted are indicated by green arrows, red arrows and red crosses, respectively. Abbreviations not already defined in the text are as follows: IPP, isopentenyl pyrophosphate; DMAPP, dimethylallyl pyrophosphate; GPPS, GPP synthase; FPS[N144W], FPP synthase N144W variant; CPR, NADPH-cytochrome P450 reductase; CYB5, cytochrome b5; GES, geraniol synthase; G8H, geraniol 8-hydroxylase; 8HGO, 8-hydroxygeraniol oxidoreductase; ISY, iridoid synthase; IO, iridoid oxidase; CYPADH, alcohol dehydrogenase 2; 7DLGT, 7-deoxyloganetic acid glucosyl transferase; 7DLH, 7-deoxyloganic acid hydroxylase; LAMT, loganic acid O-methyltransferase; TDC, tryptophan decarboxylase; GS, geissoschizine synthase; GO, geissoschizine oxidase; Redox1, protein redox 1; Redox2, protein redox 2; SAT,

stemmadenine-O-acetyltransferase; CS, catharanthine synthase; TS, tabersonine synthase; T16H, tabersonine 16-hydroxylase; 16OMT, tabersonine 16-O-methyltransferase; T3O, tabersonine 3-oxygenase; T3R, 16-methoxy-2,3-dihydro-3-hydroxytabersonine synthase; NMT, 3-hydroxy-16-methoxy-2,3-dihydrotabersonine-N-methyltransferase; D4H, deacetoxyvindoline 4-hydroxylase; DAT, deacetylvindoline-O-acetyltransferase; PRX1, class III peroxidase. The coloured boxes with dashed lines indicate strictosidine (blue), tabersonine/catharanthine (yellow) and vindoline (red) modules. The overlaps between modules are marked as green (overlap between the strictosidine and tabersonine/catharanthine modules) and orange (overlap between the tabersonine/catharanthine and vindoline modules). The information for all genes is also listed in Supplementary Table 1.

## Biosynthesis of vinblastine

The 31-step vinblastine pathway is complex, including dual inlet precursor supplies from tryptophan and geranyl pyrophosphate (GPP) as well as divergence between catharanthine and vindoline supplies

towards vinblastine biosynthesis[20,21] (Fig. 1). In *C. roseus*, the enzymes are localized in at least five compartments, namely the cytoplasm, plastid (geraniol synthase)[22], endoplasmic reticulum (cytochrome P450s and their reductase), nucleus (strictosidine-O-β-D-glucosidase, SGD)[23] and vacuole (strictosidine synthase, STR)[23], thus involving trafficking

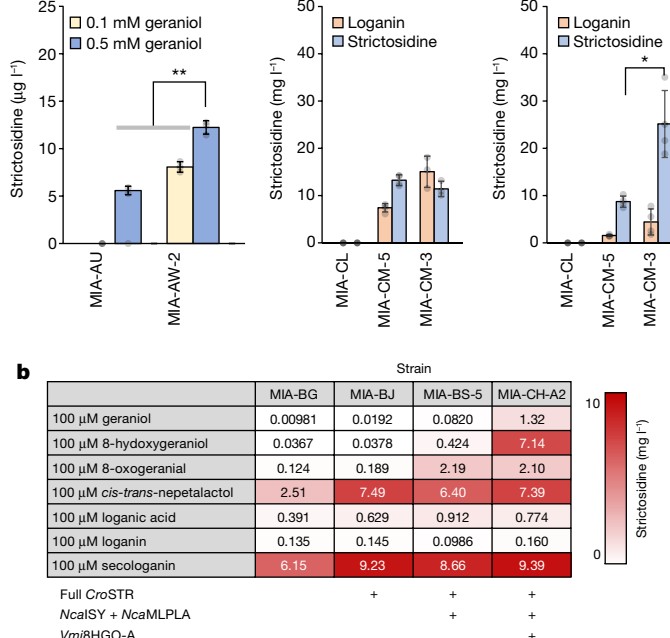

**a**

Strictosidine (μg l⁻¹) — y-axis 0–25

Legend: 0.1 mM geraniol, 0.5 mM geraniol

Strains: MIA-AU, MIA-AW-2

**

**c** SC

Strictosidine (mg l⁻¹) — y-axis 0–50

Legend: Loganin, Strictosidine

Strains: MIA-CL, MIA-CM-5, MIA-CM-3

**d** YPD

Strictosidine (mg l⁻¹) — y-axis 0–50

Legend: Loganin, Strictosidine

Strains: MIA-CL, MIA-CM-5, MIA-CM-3

*

**b**

| | Strain | | | | |
|---|---|---|---|---|---|
| | MIA-BG | MIA-BJ | MIA-BS-5 | MIA-CH-A2 | Strictosidine (mg l⁻¹) |
| 100 μM geraniol | 0.00981 | 0.0192 | 0.0820 | 1.32 | 10 |
| 100 μM 8-hydroxygeraniol | 0.0367 | 0.0378 | 0.424 | 7.14 | |
| 100 μM 8-oxogeranial | 0.124 | 0.189 | 2.19 | 2.10 | |
| 100 μM cis-trans-nepetalactol | 2.51 | 7.49 | 6.40 | 7.39 | |
| 100 μM loganic acid | 0.391 | 0.629 | 0.912 | 0.774 | |
| 100 μM loganin | 0.135 | 0.145 | 0.0986 | 0.160 | |
| 100 μM secologanin | 6.15 | 9.23 | 8.66 | 9.39 | 0 |
| Full CroSTR | | + | + | + | |
| NcaISY + NcaMLPLA | | | + | + | |
| Vm8HGO-A | | | | + | |

**Fig. 2 | Engineering and optimization of the strictosidine platform strain.**
**a**, Strictosidine production by strains MIA-AU and MIA-AW-2 grown in SC medium supplemented with strictosidine pathway precursor geraniol. **b**, Strictosidine production by engineered strains grown in SC medium supplemented with different intermediates of the strictosidine pathway. **c,d**, De novo strictosidine production from strains grown in SC (**c**) and YPD (**d**) medium. Data are presented as mean ± s.d. ($n$ = 3–4) (**a,c,d**). *$P$ value 0.05; **$P$ < 0.01. Student's two-tailed $t$-test. More statistical analysis is available in the source data file.

---

between different tissues and cell types[24]. To refactor the vinblastine pathway in yeast, we divided the *C. roseus* pathway into three modules on the basis of substrate/product availability: a strictosidine module including G8H, 8HGO, iridoid synthase, iridoid oxidase, alcohol dehydrogenase 2, 7DLGT, 7DLH, loganic acid *O*-methyltransferase, secologanin synthase (SLS) and STR[4]; a tabersonine/catharanthine module including STR (to enable strictosidine production from secologanin and tryptamine feeding), SGD, geissoschizine synthase, geissoschizine oxidase, Redox1, Redox2, SAT, *O*-acetylstemmadenine oxidase precondylocarpine acetate synthase (PAS), DPAS (dihydroprecondylocarpine acetate synthase), catharanthine synthase and tabersonine synthase[20] and a vinblastine module including T16H1, T16H2, 16OMT, T3O, T3R, NMT, D4H, deacetylvindoline-*O*-acetyltransferase[17] and PRX1 (ref. [25]) (Fig. 1 and Supplementary Table 1). Furthermore, each module included the expression of *C. roseus* cytochrome P450 reductase (*Cro*CPR) as well as cytochrome b5 (*Cro*CYB5) to support the cytochrome P450 enzymes.

## A de novo strictosidine platform in yeast

The initial strictosidine strain MIA-AU was designed on the basis of a previously reported 12-step de novo strictosidine strain, including deletion of *ATF1* and *OYE2*, encoding an alcohol acetyltransferase and a reduced nicotinamide adenine dinucleotide phosphate (NADPH) oxidoreductase, respectively, to minimize by-product formation from geraniol[4] (Fig. 1). In addition, MIA-AU was designed to down-regulate *ERG20* encoding farnesyl pyrophosphate (FPP) synthase to improve the GPP:FPP ratio. However, when tested in the synthetic complete (SC) medium with all amino acids and 2% glucose, MIA-AU did not produce detectable amounts of strictosidine. Yet, when supplementing 500 μM geraniol to the medium (no higher concentration was tested owing to high toxicity of geraniol above 500 μM, Extended Data Fig. 1), MIA-AU produced 5.59 μg l⁻¹ (10.5 nM) strictosidine, indicating

geraniol precursor supply limitations (Fig. 2a). We then deleted the NADPH-dependent aldehyde reductase *ARI1* and NADPH oxidoreductase *OYE3*, involved in syphoning 8-hydroxygeraniol and 8-oxogeranial, respectively, to shunt products[26], resulting in strain MIA-AW-2 (Supplementary Table 2). When grown in SC medium supplemented with 500 μM geraniol, MIA-AW-2 produced 12.2 μg l⁻¹ (23.1 nM) strictosidine (Fig. 2a).

Acknowledging the challenges often encountered when expressing plant cytochrome P450s in microbes[27], we first investigated the expression and subcellular localization of the four P450 enzymes of the strictosidine module, G8H, iridoid oxidase, 7DLH and SLS (Supplementary Fig. 1), each fused to a yeast enhanced green fluorescent protein (yEGFP). Although the fluorescence was generally low, we observed that all four P450 enzymes had correct endoplasmic reticulum membrane localization (Supplementary Fig. 1). Using targeted proteomics, we also observed poor expression of GPP synthase, FPS and geraniol synthase, which are responsible for geraniol production, as well as iridoid oxidase and SLS (Extended Data Fig. 2b), consistent with the fluorescence microscopy result, altogether suggesting several inefficient steps requiring further optimization.

To enable iterative engineering and refactoring of long biosynthetic pathways, we included a set of unique 20-nucleotide guide-RNA sequences in front of the promoter of each expression cassette, allowing for easy swapping of target promoters or entire expression cassettes iteratively in strain MIA-BG (Methods and Extended Data Fig. 3a). Like MIA-AU, the new platform strain MIA-BG produced 9.81 μg l⁻¹ (18.5 nM) strictosidine when fed 100 μM geraniol and tryptamine. We supplemented the cultivation medium with tryptamine in addition to one of the precursors 8-hydroxygeraniol, 8-oxogeranial, *cis-trans*-nepetalactol, loganic acid, loganin or secologanin to deduce possible pathway bottlenecks. Here, when feeding 8-hydroxygeraniol, MIA-BG produced 36.7 μg l⁻¹ (69.1 nM) strictosidine, whereas feeding 8-oxogeranial or *cis-trans*-nepetalactol resulted in 124 μg l⁻¹ (233 nM) or 2.51 mg l⁻¹ (4.73 μM) strictosidine, respectively, indicating 8HGO and iridoid synthase as the main bottlenecks. Furthermore, the highest strictosidine production in MIA-BG (6.15 mg l⁻¹ or 11.6 μM) was achieved by feeding 100 μM secologanin (Fig. 2b and Extended Data Fig. 4). However, the unused secologanin in the spent medium further suggested STR as a rate-limiting step for strictosidine production, which could be partially alleviated by the overexpression of t*Cro*STR from a high copy plasmid, resulting in a substantial improvement of strictosidine titre to 39.1 mg l⁻¹ (73.7 μM), whereas 8.53 mg l⁻¹ (21.9 μM) secologanin remained in the medium (Supplementary Fig. 2).

To mitigate the bottlenecks in strictosidine production, we focused first on the Pictet–Spengler-type reaction catalysed by STR by screening seven STR homologues identified from MIA-producing plants in both full-length and truncated versions (tSTR) without the N-terminal vacuolar signal peptide[4]. Individually, these variants were introduced into a strain without STR (MIA-BC), and the resulting strains were cultivated in the presence of secologanin and tryptamine. The result showed that full-length STRs yielded 2–3 times more strictosidine compared to tSTRs, with *C. roseus* STR (*Cro*STR) enabling the highest titre of 17.5 mg l⁻¹ (32.9 μM) strictosidine (Extended Data Fig. 5a). Fluorescence microscopy analysis showed that *Cro*STR was localized in the vacuole or endoplasmic reticulum membrane, while the truncated variant was localized to the nucleus in fresh cultures and shifted to the cytoplasm following overnight growth (Supplementary Fig. 3), suggesting that either the substrates secologanin and tryptamine are mostly in the vacuole, or the acidic pH in the vacuole is favourable for strictosidine synthesis. Alternatively, post-translational modifications or specific folding steps encountered along the endoplasmic reticulum-to-vacuole trafficking pathway may support STR activity. Swapping the t*Cro*STR in MIA-BG with *Cro*STR (resulting in strain MIA-BJ) enabled an efficient (80%) conversion of 50 μM secologanin to 21.2 mg l⁻¹ (40.0 μM) strictosidine, a 6.9-fold improvement over t*Cro*STR (Extended Data Fig. 5b).

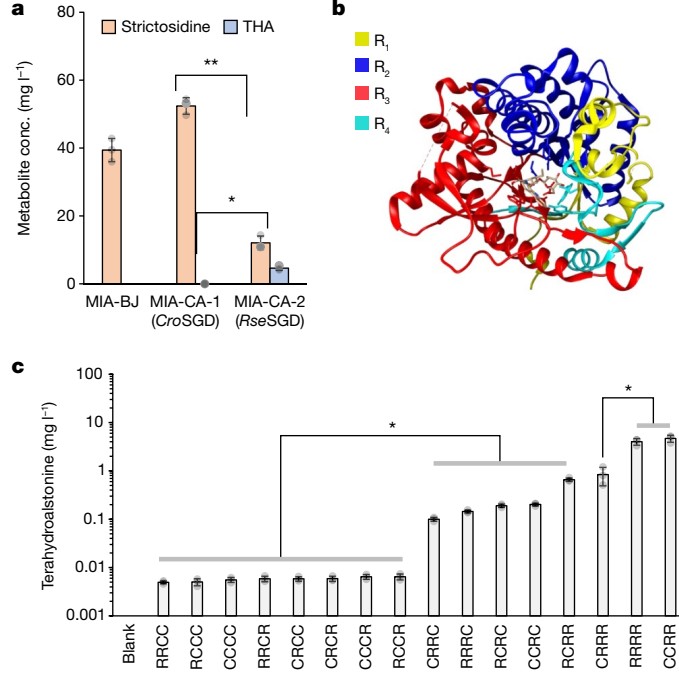

**a** (chart: Metabolite conc. (mg l⁻¹) vs MIA-BJ, MIA-CA-1 (*Cro*SGD), MIA-CA-2 (*Rse*SGD); Strictosidine and THA)

**b** R₁ (yellow), R₂ (blue), R₃ (red), R₄ (cyan)

**c** (chart: Tetrahydroalstonine (mg l⁻¹) vs Hybrid SGD: Blank, RRCC, RCCC, CCCC, RRCR, CRCC, CRCR, CCCR, RCCR, CRRC, RRRC, RCRC, CCRC, RCRR, CRRR, RRRR, CCRR)

**Fig. 3 | Functionalization of strictosidine-β-D-glucosidase (SGD) in yeast.**
**a**, THA production in yeast strains expressing *Cro*STR and *Cro*THAS together with SGD from *C. roseus* (*Cro*SGD) or *R. serpentina* (*Rse*SGD). conc., concentration.
**b**, *Rse*SGD protein divided into four domains on the basis of sequence conservation between *Cro*SGD and *Rse*SGD, denoted as R₁ (yellow), R₂ (blue), R₃ (red) and R₄ (cyan); crystal structure (PDB ID 2jf6). **c**, THA production from hybrid SGDs constructed by shuffling four domains between *C. roseus* (indicated by C) and *R. serpentina* (indicated by R) sequences. The first letter of the hybrid SGDs on the x axis is domain 1, the second letter domain 2 and so on. Data are presented as mean ± s.d. ($n = 3$) (**a**,**c**). $*P < 0.01$; $**P < 0.0001$. Student's two-tailed *t*-test. More statistical analysis is available in the source data file.

Next, to optimize the upstream steps of the strictosidine module, we screened a combinatorial library of four iridoid synthases and four cyclases (MLPs and NEPS2), which are critical enzymes involved in the cyclization of 8-oxogeranial to nepetalactol[28,29], in strain MIA-BKV-1 (deleting *Cro*ISY from MIA-BJ) (Supplementary Table 2). When fed 100 μM 8-oxogeranial and tryptamine, strains expressing *Nca*ML-PLA in combination with *Nca*ISY reached the highest titres among the tested combinations, producing 2.18 mg l⁻¹ (5.79 μM) loganic acid, 12.0 mg l⁻¹ (30.8 μM) loganin, 1.20 mg l⁻¹ (3.09 μM) secologanin and 1.20 mg l⁻¹ (2.26 μM) strictosidine (42.0 μM combined) (Extended Data Fig. 6). Both *Nca*ISY and *Nca*MLPLA were integrated into strain MIA-BKV-1, resulting in strain MIA-BS-5 (Supplementary Table 2). Similarly, to optimize 8HGO we screened a combinatorial library of two NAD⁺-dependent (type-A) and four NADP⁺-dependent (type-B) 8HGOs in MIA-BKV-5 (deleting *Cro*8HGO-B from the strain MIA-BS-5). Feeding 100 μM 8-hydroxygeraniol and tryptamine to these strains we identified *Vinca minor* 8HGO-A (*Vmi*8HGO-A) as the most efficient enzyme supporting a production of 19.9 μg l⁻¹ (248 nM) loganic acid, 20.7 μg l⁻¹ (8.35 μM) loganin, 20.6 μg l⁻¹ (4.07 μM) secologanin and 28.1 μg l⁻¹ (3.64 μM) strictosidine (or 16.3 μM combined). Empty-vector control strains produced 1.86 mg l⁻¹ (4.38 μM) of the same metabolites, indicating native yeast oxidoreductase activity on 8-hydroxygeraniol. Coexpression of 8HGO from either *Rauvolfia tetraphylla* (*Rte*8HGO-B) or *Sesamum indicum* (*Sin*8HGO-B), together with *Vmi*8HGO-A, had an insignificant effect, whereas coexpression of *C. roseus* (*Cro*8HGO-B) or *Populus trichocarpa* (*Ptr*8HGO-B) had a negative effect on the production, probably because *Cro*8HGO-B and *Ptr*8HGO-B catalysed the

reaction in the reverse direction (Extended Data Fig. 7). Integrating *Vmi*8HGO-A into MIA-BKV-5 generated strain MIA-CH-A2 (Supplementary Table 2), which produced 1.32 mg l⁻¹ (2.49 μM) strictosidine when supplemented with 100 μM geraniol and tryptamine (Fig. 2b).

Last, to produce strictosidine de novo from glucose and tryptophan, we increased geraniol availability by dynamically down-regulating *ERG9* and *ERG20* (both encoding GPP consuming enzymes) using the glucose-dependent P_{HXT1} and P_{HXT3} promoters, respectively[30], resulting in strain MIA-CL (Supplementary Table 2). We further overexpressed a specific GPP synthase encoded by *Agr*GPPS2 (ref. [31]). *Gga*FPS^{N144W}, which is an FPP synthase that has lower affinity for GPP[32] and an *ERG20* variant (*ERG20*^{F96W,N127W}) that shows a low FPP synthase activity[33]. We fused *ERG20*^{F96W,N127W} to t*Cro*GES, to enhance the GPP pool and redirect it towards the first precursor, geraniol. *IDI1* and truncated *HMG1* (t*HMG1*) were overexpressed to increase the synthesis of isopentenyl pyrophosphate and dimethylallyl pyrophosphate, both precursors for GPP[4,34]. Finally, *Cro*TDC was overexpressed to convert tryptophan to tryptamine, resulting in strain MIA-CM-5 (Supplementary Table 2). When grown in SC medium, strain MIA-CM-5 produced 7.43 mg l⁻¹ (19.0 μM) loganin and 13.3 mg l⁻¹ (25.0 μM) strictosidine (Fig. 2c). As glucose is depleted in 1–2 days during the 6-day cultivation, and yeast continues to use ethanol produced from glucose, we replaced the constitutively active P_{TEF2} and P_{CCW12} promoters driving expression of genes encoding *Agr*GPPS2 and *ERG20*^{F96W,N127W}-t*Cro*GES fusion with glucose-repressible promoters P_{MLS1} and P_{ICL1} to avoid competition for GPP during growth on glucose and to ensure high expression on ethanol after glucose was depleted. In SC medium, MIA-CM-3 produced 15.0 mg l⁻¹ (38.6 μM) loganin and 11.4 mg l⁻¹ (21.5 μM) strictosidine, concentrations similar to those produced by MIA-CM-5 (Fig. 2c). However, when grown in yeast extract peptone dextrose (YPD) medium, MIA-CM-3 reached 25.2 mg l⁻¹ (47.4 μM) strictosidine, which was considerably higher than MIA-CM-5 (8.73 mg l⁻¹ or 16.5 μM), and both MIA-CM-5 and MIA-CM-3 strains produced notably lower loganin in YPD medium compared to SC medium (Fig. 2c–d and Supplementary Fig. 4).

## Engineering of the SGD gateway

Having optimized de novo strictosidine production, we next sought to express the gateway enzyme strictosidine-β-D-glucosidase (SGD), together with one-step conversion of strictosidine aglycone to tetrahydroalstonine (THA), which is not part of the main vinblastine pathway but is the shortest downstream pathway to a side product for which a chemical standard is available[35] (Extended Data Fig. 8a). Despite extensive studies on SGD, including a crystal structure of the *Rauvolfia serpentina* SGD (*Rse*SGD) and functional expression of *Cro*SGD in yeast[36], we initially did not detect any strictosidine aglycone when expressing *Cro*SGD alone, or THA when expressing *Cro*SGD together with *C. roseus* THA synthase (*Cro*THAS) in strain MIA-BJ (Fig. 3a and Supplementary Fig. 5). Several attempts to express full-length or truncated CroTHAS (removing the nuclear localization sequence K_{214}K_{215}K_{216}R_{217}) fused to full-length or truncated *Cro*SGD (removing the C-terminal nuclear localization sequence K_{537}KRFREEDKLVELVKKQKY_{555}) did not yield detectable THA production in yeast (data not shown). However, when testing 46 homologues of *Cro*SGD, *Rse*SGD yielded 4.02 mg l⁻¹ (11.4 μM) THA from feeding 100 μM secologanin and 1 mM tryptamine, and another 15 homologues yielded more than 0.1 μM or 35.2 μg l⁻¹ (0.1% yield from 100 μM secologanin feeding) THA (Extended Data Fig. 8b). Despite *Cro*SGD being previously reported to localize to the nucleus in plant cells[35], fluorescence microscopy of yeast cells expressing N-terminally yEGFP-tagged *Cro*SGD and *Rse*SGD showed different subcellular localization, with *Rse*SGD localizing in the nucleus and *Cro*SGD showing diffuse cytoplasmic distribution (Supplementary Fig. 6). As SGD is reported to form a protein complex in the nucleus in plant cells[35], the cytoplasmic location of *Cro*SGD was presumably the main reason for the lack of activity from *Cro*SGD. The distinct protein

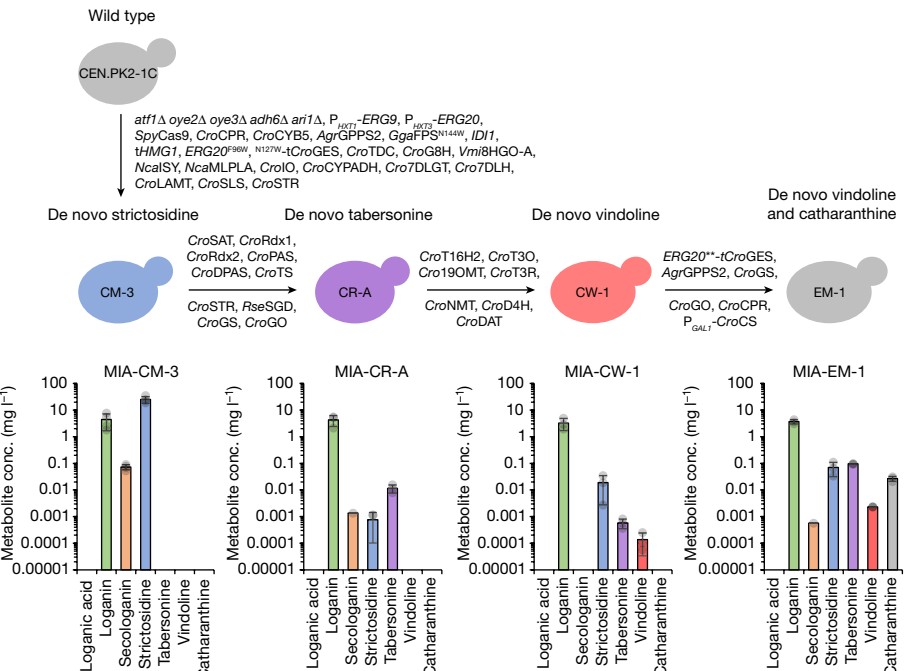

**Fig. 4 | De novo synthesis of strictosidine, tabersonine, vindoline and catharanthine in engineered yeast strains.** Strains MIA-CM-3 and MIA-CR-A were grown in 96-well deep plates containing 0.5 ml of SC medium. Strain MIA-CW-1 was grown in a 24-well deep plate (2 ml). Strain MIA-EM-1 was grown in a 48-well BioLector Pro flower-shaped plate (1 ml) with continuous feeding of glucose (fed-batch). Data represented are mean ± s.d. (*n* = 3–4). *C. roseus* catharanthine synthase, *Cro*CS.

localization/activity was surprising given the 70% protein identity between these two homologues. To investigate sequence or structural features indispensable for *Rse*SGD function in yeast, we divided SGD into four domains on the basis of sequence conservation between *Rse*SGD and *Cro*SGD (Fig. 3c and Supplementary Table 3), and tested THA production from all possible 16 hybrid SGDs constructed from the four domains of the two SGDs. From this, we observed that the third domain from *Rse*SGD ($R_3$), is required for SGD activity while maintaining the integrity of the third and fourth domains ($R_3+R_4$) produced the highest amount of THA (4.69 mg l$^{-1}$ or 13.3 µM) (Fig. 3c), probably contributing to SGD complex formation or correct localization in the nucleus (Supplementary Fig. 6).

### Functional expression of MIA modules

Next, to refactor the tabersonine/catharanthine module from *C. roseus* in yeast, we first constructed strain MIA-DC by integrating all ten genes from STR (to enable strictosidine production from fed secologanin and tryptamine) to tabersonine and catharanthine synthases together with *Cro*CPR and *Cro*CYB5 for the cytochrome P450 enzyme geissoschizine oxidase[20,21] (Fig. 1 and Extended Data Fig. 9). When fused to a C-terminally tagged yEGFP, all proteins, except for geissoschizine synthase and PAS, showed relatively strong fluorescence (Supplementary Fig. 7). When fed 100 µM secologanin and 1 mM tryptamine, strain MIA-DC produced 9.29 µg l$^{-1}$ (27.6 nM) tabersonine and 6.96 µg l$^{-1}$ (20.7 nM) catharanthine (Extended Data Fig. 9).

In parallel, we refactored the vinblastine module (strain MIA-DE) by integrating all nine genes from tabersonine 16-hydroxylase (encoded by T16H1 and T16H2) to peroxidase PRX1, which was suggested to condense vindoline and catharanthine, together with *Cro*CPR and *Cro*CYB5, into a wild-type strain[17,19] (Fig. 1 and Extended Data Fig. 9). When fed 100 µM tabersonine and 100 µM catharanthine, strain MIA-DE produced 0.992 mg l$^{-1}$ (2.17 µM) vindoline but no detectable vinblastine (Extended Data Fig. 9). Targeted metabolite analysis identified a strong signal for masses identical to 16-hydroxytabersonine

and 16-methoxytabersonine, indicating a potential bottleneck at the P450 enzyme T3O (Extended Data Fig. 9). This was consistent with the improved vindoline production when extra copies of two consecutive genes in the vinblastine module were expressed in the MIA-DE strain, albeit with overexpression of T16H2 and 16OMT yielding the highest improvement (more than twofold) in vindoline titre compared to strain MIA-DE (Extended Data Fig. 9). Whereas strain MIA-DE did not produce vinblastine ($C_{46}H_{58}N_4O_9$) from vindoline and catharanthine feeding, high-resolution MS detected a compound with formula $C_{46}H_{56}N_4O_9$ in both wild-type and MIA-DE strains, suggesting non-enzymatic leurosine formation as also previously shown to be synthesized in vitro from catharanthine and vindoline[37,38]. Confocal microscopy analysis indicated that *Cro*PRX1 was not expressed/folded properly (Supplementary Fig. 8), and several attempts to truncate PRX1 at the N and/or C termini did not yield any vinblastine production when cells were grown in media supplemented with 100 µM catharanthine and 100 µM vindoline. Integrating the vinblastine module into strain MIA-DC (catharanthine-tabersonine strain) yielded strain MIA-DJ, which produced 44.0 µg l$^{-1}$ (130 nM) catharanthine and 4.37 µg l$^{-1}$ (9.57 nM) vindoline when cultivated in YPD medium supplemented with 100 µM secologanin and 1 mM tryptamine, whereas strain MIA-DC produced 11.4 µg l$^{-1}$ (34.0 nM) tabersonine and 90.8 µg l$^{-1}$ (270 nM) catharanthine, indicative of catharanthine synthase having better expression or higher catalytic reactivity than tabersonine synthase (Extended Data Fig. 9). These results are in line with untargeted proteomics data showing strain MIA-DJ had relatively low expression of tabersonine synthase (Extended Data Fig. 10).

### De novo vinblastine precursor synthesis

To make tabersonine and catharanthine de novo from glucose and tryptophan, we integrated the entire tabersonine module, including *Rse*SGD, and *C. roseus* geissoschizine synthase, geissoschizine oxidase, Redox1, Redox2, SAT, PAS, DPAS, tabersonine synthase (leaving out catharanthine synthase as it competes for dihydroprecondylocarpine

acetate, the precursor for tabersonine synthase), into the best-performing de novo strictosidine production strain MIA-CM-3 (Fig. 2d). The resulting strain MIA-CR-A produced 11.5 µg l⁻¹ (34.2 nM) tabersonine from YPD with 2% glucose as the only carbon source (Fig. 4b). We further integrated the seven vindoline biosynthetic genes (*C. roseus* T16H2, 16OMT, T3O, T3R, NMT, D4H and DAT)[17,19] into strain MIA-CR-A. The resulting strain, designated MIA-CW-1, produced 0.137 µg l⁻¹ (0.307 nM) vindoline in YPD medium with 2% glucose (Fig. 4b).

To enable one-pot de novo production of vindoline and catharanthine, *C. roseus* catharanthine synthase was integrated into strain MIA-CW-1 under the control of a galactose inducible ($P_{GAL1}$) promoter. Extra copies of the *ERG20*\*\*-t*Cro*GES fusion and *Agr*GPPS2 were integrated to boost geraniol synthesis. To enhance the conversion of the unstable and highly reactive intermediate strictosidine aglycone[23] to geissoschizine and its oxidation, we also integrated an extra copy of *Cro*GS, *Cro*GO and *Cro*CPR, resulting in the strain MIA-EM-1. Using biphasic fermentation with glucose and galactose as carbon sources, MIA-EM-1 produced 2.32 µg l⁻¹ (5.08 nM) vindoline and 26.6 µg l⁻¹ (79.1 nM) catharanthine de novo (Fig. 4 and Extended Data Fig. 10). We further constructed strain MIA-EM-2 by integrating an extra copy of each of the six tabersonine module genes, namely *Cro*GS, *Cro*GO, *Cro*SAT, *Cro*PAS, *Cro*DPAS and *Cro*TS, into strain MIA-EM-1. Similarly, we constructed MIA-EM-3 by integrating an extra copy of *Cro*TS as well as the vindoline module genes (*Cro*T16H2, *Cro*T3O, *Cro*T3R, *Cro*D4H and *Cro*DAT) (Extended Data Fig. 10). Under the same fed-batch cultivation condition, strain MIA-EM-2, which had an enhanced tabersonine module, produced 5.48 µg l⁻¹ (12.0 nM) vindoline and 69.4 µg l⁻¹ (206 nM) catharanthine (Extended Data Fig. 10). On the other hand, strain MIA-EM-3 (with a further copy of vindoline module genes) produced 4.19 µg l⁻¹ (9.18 nM) vindoline and only one out of three colonies produced 6.46 µg l⁻¹ (19.2 nM) catharanthine (Extended Data Fig. 10), indicating that the main bottleneck was still within the tabersonine/catharanthine module and further enhancing the vindoline module expression was detrimental to catharanthine production because of a stronger pull of dihydroprecondylocarpine acetate, which is also a precursor to catharanthine (Fig. 1). Subsequently, strain MIA-EM-2 was grown in fed-batch cultivation in ambr 250 bioreactors, with continuous feeding of 3× SC containing 2% glucose and 3 mM tryptophan (between 20 and 72 h), and daily dosing of 3 g l⁻¹ galactose during days 4–10. Samples were collected every 24 h to analyse the MIA metabolites. At the end of the fermentation (day 11), the catharanthine and vindoline titres reached 91.4 µg l⁻¹ (271 nM) and 13.2 µg l⁻¹ (29.0 nM), respectively (Extended Data Fig. 11 and Supplementary Fig. 9).

## Vinblastine semisynthesis

Vindoline and catharanthine can be enzymatically coupled using a peroxidase and peroxide as demonstrated previously[25], chemically coupled using Fe(III)[2] or photo-chemically coupled using near-UV radiation in the presence of flavin mononucleotide[38]. The enzymatic coupling using horseradish peroxidase did not yield a detectable amount of vinblastine (data not shown), and we could only produce 336 µg l⁻¹ (414 nM) and 397 µg l⁻¹ (490 nM) vinblastine from pure catharanthine (100 µM) and vindoline (10 µM) standards using the Fe(III)-based coupling and photo-chemical methods, respectively (Extended Data Fig. 12). Also, attempts to couple yeast-produced catharanthine and vindoline (after liquid–liquid extraction and roughly 750-fold concentration of the compounds, see Supplementary Methods) using the Fe(III) method was unsuccessful (data not shown), presumably because of unspecific coupling to by-products or a strong suppression of vinblastine signal by the complex matrix concentrated from the fermentation broth. However, following purification using thin-layer chromatography and subsequent preparative high-performance liquid chromatography (HPLC) (Supplementary Methods), we could detect 23.9 µg l⁻¹ (29.4 nM) vinblastine using the Fe(III) coupling method (Extended Data Fig. 12).

A considerable amount of the intermediate anhydrovinblastine was also formed yet was not quantifiable owing to a lack of an analytical standard.

## Discussion

Here we report the optimization of the strictosidine biosynthetic pathway by more than 1,000-fold, reaching titres of 25.2 mg l⁻¹ (47.4 µM) in 96-well plates (Fig. 2d). From this, we demonstrated de novo production of catharanthine and vindoline, with the 30-step vindoline pathway representing a very long heterologous pathway refactored in a microbial cell. Furthermore, and in alignment with industrial scale manufacturing of vinblastine, catharanthine and vindoline were purified from fermentation broths and chemically coupled to form vinblastine (Extended Data Fig. 12). Although we were unable to make vinblastine directly in yeast (as *Cro*PRX1 proved difficult to express in yeast as indicated by proteomics analysis of the MIA-DJ strain), our microbial process mimics the plant-based production process, which relies on coupling vindoline and catharanthine after their purification from *C. roseus* leaves. Although it may one day be possible to identify a PRX1 homologue that successfully catalyses the reaction when compartmentalized in the peroxisome, nevertheless this study provides a demonstration of a microbial supply chain for a complex essential anti-cancer medicine. Microbial supply chains for essential medicines with few other suppliers, long lead times and high manufacturing complexity, are viable solutions to secure the global or regional supply of these drugs, especially during unpredictable environmental events, such as natural disaster, plant disease and pandemics[39]. This said, further metabolic engineering and bioprocess optimization are needed for engineered yeast to supplement and ultimately stabilize existing *C. roseus*-based vinblastine supply chains. Specifically, the tabersonine/catharanthine module, which had many promiscuous reactions, potential host–heterologous pathway interactions and protein aggregation, was particularly challenging[20]. Furthermore, expression of the seven P450 enzymes could be further optimized, or they could be replaced with better homologues. Likewise, de novo vindoline production required more deletion, knockdown and overexpression of ten native yeast genes to improve precursor supplies, which also need to be regulated to ensure balanced cell growth and production in future process optimizations. Having said this, we consider both the strictosidine platform, and the demonstration of one-pot de novo vindoline and catharanthine production ripe for new MIA pathway discoveries as well as production and bioactivity testing of new unnatural MIA analogues with higher potency and/or efficacy, such as 10-fluoro-vinblastine[40], in yeast cell factories.

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

## Methods

### Chemical standards

All chemical standards (Supplementary Table 2) had a purity of 95% or higher.

### Gene discovery

Sequence for *Vmi*8HGO-A was obtained from *V. minor* transcriptome data[41] using BLASTn[42] and the *C. roseus* 8HGO-A sequence (GenBank accession no. AHK60836.1) as the query. Open reading frames were predicted using TransDecoder (http://transdecoder.github.io).

### Plasmid construction

All plasmids were constructed by USER cloning[43] or restriction enzyme cloning, followed by chemical transformation into *E. coli* DH5α competent cells, and plated on Luria–Bertani agar containing 100 μg ml$^{-1}$ ampicillin and grown at 37 °C overnight (16–20 h). *E. coli* transformants were genotyped by colony PCR, and colonies with correct size of insertion were grown in 5 ml of Luria–Bertani medium containing 100 μg ml$^{-1}$ ampicillin at 37 °C for 16–20 h, followed by miniprep plasmid extraction and validated by Sanger sequencing. All biosynthetic genes (Supplementary Table 1) were codon optimized for yeast expression and synthesized by either IDT or Twist Biosciences. All genes were assembled as expression cassettes in either centromeric (pRS series) or 2-μm (pESC series) plasmids by USER cloning. Single guide-RNA plasmids were constructed by USER cloning a double-stranded oligo containing the 20-nucleotide guide-RNA sequence into a 2-μm vector with either the *LEU2* or *URA3* selection marker. Several guide-RNA plasmids were constructed by USER assembly of all guide-RNA expression cassettes amplified from single guide-RNA plasmids. All plasmids (Supplementary Table 4) were verified by Sanger sequencing of the insertion.

### Yeast strain construction

All yeast strains (Supplementary Table 2) were constructed on the basis of the wild-type strain CEN.PK2-1C using the lithium acetate/single-stranded carrier DNA/PEG protocol[44], and have the same auxotrophy for uracil, histidine, leucine and tryptophan. Genome editing (integration, deletion or swapping) was performed using the CasEMBLR method[45]. All DNA parts were amplified from precloned plasmids using Phusion High-Fidelity PCR Master Mix with HF Buffer (Thermo Fisher Scientific) using primers containing 25–30 bp overhangs (IDT). For strains MIA-AU and MIA-AW-2, *Spy*Cas9 was expressed from a centromeric plasmid with the *TRP1* selection marker. For all other MIA strains, CEN.PK2-1C wild-type strain was first transformed with a plasmid (*TRP1*) harbouring the *Spy*Cas9 cassette, followed by another transformation to integrate the *Spy*Cas9 cassette into EasyClone site XII-1 for stable expression[46]. For all editing were transformed with a plasmid expressing a single or multiple-targeting guide-RNA fragments, as well as DNA parts as repair templates Transformants were plated on SC agar lacking the appropriate amino acid(s) (uracil, leucine, histidine or tryptophan) for selection of the guide-RNA plasmid, and grown for 3–4 days at 30 °C. All strains with genomic integration were confirmed by genotyping PCR and Sanger sequencing of the integrated gene cassettes.

For all strictosidine strains except MIA-AU and MIA-AW-2, a unique guide-RNA sequence was included at the beginning of all gene cassettes (beginning of the promoter sequence) for easy swapping of the promoter or deletion of whole gene cassettes (Extended Data Fig. 3a). To swap a promoter/gene in an expression cassette, yeast cells were transformed with a plasmid expressing the guide-RNA, together with a new promoter/gene and parts overlapping with sequences up- and downstream of this promoter/gene (Extended Data Fig. 3b). Similarly, to delete a whole cassette, yeast cells were transformed with a plasmid expressing the specific guide-RNA and two parts (with a 50–60 bp overlapping sequence) that are homologous to the up- and downstream of the locus to be deleted (Extended Data Fig. 3c). Transformants were plated on SC agar lacking either leucine or uracil for selection of the guide-RNA plasmid and confirmed by colony PCR and sequencing.

### Media and yeast cultivation

Yeast cells were cultivated in either SC defined medium or YPD rich medium, both containing 2% glucose as the carbon source (except for the screening of the iridoid synthase + cyclase and 8HGO variants, which used glucose and galactose). For batch-fed cultivations, 3× SC medium (2% glucose) supplemented with 3 mM tryptophan was used to reach a high biomass.

For all small-scale cultivations, precultures were typically made by growing yeast colonies (biological replicates) in 150 μl of SC medium (lacking appropriate amino acid for auxotrophic selection) or YPD medium on a 96-well microtitre plate and incubated at 30 °C shaking at 300 r.p.m. overnight (roughly 16–20 h). The maximum specific growth rate for all yeast strains were characterized using the Growth Profiler 960 system (Enzyscreen) (Supplementary Fig. 9). All cultures were grown in 300 μl of SC medium in 96-well square-shaped microtitre plates for at least 3 days. Cultures were grown at 30 °C with shaking speed of 250 r.p.m. and the cell density was measured every 30 min. Data from the exponential growth phase were used to determine the maximum specific growth rate by linear regression. Small-scale production assays were performed using either 96-well (with 0.5 ml of medium) or 24-well (with 2 ml of medium) deep-well plates. After 1 day (typically reaching the stationary phase), 10 μl of precultures were transferred to either SC lacking appropriate amino acid(s) or YPD medium supplemented with one or more precursors. For testing strictosidine production, yeast strains were fed with 0.1–1 mM tryptamine and one of the precursors including geraniol, 8-hydroxygeraniol, 8-oxogeranial, *cis-trans*-nepetalactol, loganic acid, loganin and secologanin. For testing tabersonine and catharanthine production, cells were fed with 1 mM tryptamine and 50–250 μM secologanin. For testing vindoline or vinblastine production, cells were fed with 100 μM tabersonine or 100 μM vindoline and 100 μM catharanthine, respectively. All compounds used in the feeding experiments are listed in Supplementary Table 5.

Small-scale fed-batch cultivations were performed in BioLector Pro systems using flower-shaped plates. The fermentation started with inoculating 1 ml 3× SC (2% glucose) medium supplemented with 3 mM tryptophan to an initial OD$_{600}$ of 0.5. Continuous exponential feeding with the same medium except containing 36% glucose started at 20 h using the following equation:

$$\text{Flow rate } (\mu l\ h^{-1}) = 0.48e^{0.0125t}$$

where $t$ is the feeding time (h).

Temperature was kept at 30 °C throughout the whole cultivation. The pH was controlled at 5.5 using 10% NH$_4$OH solution. The plate was shaken at speed 1,000 r.p.m. The relative humidity in the growth chamber was maintained at 85% using distilled water to minimize evaporation of the media. An automated liquid handler was used to take 0.1 ml culture samples on days 3, 5 and 7 for metabolite analysis.

Batch-fed fermentations were performed on an ambr 250 system using single-use microbial vessels. The fermentation started by inoculating 100 ml of 3× SC medium (2% glucose) supplemented with 3 mM tryptophan to an initial OD$_{600}$ of 5.0. The same medium, except containing 36% glucose was fed continuously into the bioreactor between 20 and 96 h using the following equation:

$$\text{Flow rate } (ml\ h^{-1}) = 48e^{0.0125t}$$

Between days 4 and 10, 3 g l$^{-1}$ of galactose was added into the fermentation every day to induce the expression of catharanthine synthase. The temperature was kept at 30 °C throughout the whole fermentation. The pH of all bioreactors was controlled at 5.0 using 10% NH$_4$OH solution. The air flow rate was initially at 1 volume of air per unit of medium

per unit of time (vvm), and dissolved oxygen ($dO_2$) was maintained above 40% by increasing the agitation speed. $CO_2$ in the exhaust gas was monitored by a gas analyser. An automated liquid handler was used to take 1.0 ml broth samples every 24 h for metabolite analysis. At the end of the cultivation, all (150–200 ml) broth was used for crude extraction of MIA compounds including catharanthine and vindoline.

## MIA compounds extraction and purification
In some cases, MIA metabolites were extracted from fermentation broths using the Oasis HLB (hydrophilic–lipophilic balance) cartridges and 96-well plates following the manufacturer's manual (Waters). Solid phase extraction (SPE) of all samples (0.5–2 ml) from 96- or 24-well deep plates were performed on Oasis HLB 96-Well Plates (SKU 186000679) and eluted with 0.5 ml of pure methanol. Samples taken from the ambr 250 bioreactors were extracted using dichloromethane and the resulting crude extract was further purified by preparative thin-layer chromatography followed by preparative HPLC (Supplementary Methods). The final samples were dissolved with appropriate amounts of water and used for chemical coupling to produce vinblastine.

## Synthesis of vinblastine from vindoline and catharanthine
Photo-chemical coupling of vindoline and catharanthine was performed in 0.1 M Tris-HCl buffer pH 7.0 containing 1 mM $MnCl_2$ (ref. [38]). The reaction was initiated by adding 100 µM catharanthine, 10 µM vindoline and 50 µM flavin mononucleotide in a 96-well plate. The reaction (200 µl) was then placed under UV-A light (peak wavelength at 365 nm) for 5 min. After the exposure, 20 µl of 100 mM NADH solution was added (9.1 mM final concentration) and incubated at 0 °C for 5 h. Chemical coupling of vindoline and catharanthine was performed in 75 mM $FeCl_3$ and 0.05 M HCl containing 10% (v/v) 2,2,2-trifluoroethanol and incubated at 25 °C for 15 min, followed by a reduction using 10 mM $NaBH_4$ at 0 °C for 4 h (ref. [2]). All assay samples were kept at −20 °C until SPE using an Oasis HLB 96-well plate (Waters) following the manufacturer's protocol.

## Metabolites sample preparation and analysis
After 3 or 6 days, 200 µl of cultures were mixed with 20 µl of 10 mg l$^{-1}$ caffeine solution (as the internal standard for normalization) and filtered through a filter plate (PALL, AcroPrep Advance, 0.2 µm Supor membrane for media/water) by centrifugation at 2,200$g$ for 1 min. For tabersonine and vindoline analysis, strains MIA-CR-A and MIA-CW-1 were also cultivated in 4 ml of YPD medium on a 24-well deep-well plate for 6 days, and the whole spent medium was used for SPE using Oasis HLB 96-well plate (Waters) following manufacturer's protocol. Metabolites were eluted with 0.8 ml of pure methanol, evaporated to dryness at room temperature and reconstituted in 100 µl of water and mixed with 10 µl of 10 mg l$^{-1}$ caffeine solution before filtration. A series of standard mixtures (Supplementary Table 5) were prepared in the same way as analytical samples. Typically, 1 µl samples were injected for analysis. For de novo tabersonine, catharanthine and vindoline strains, 3 µl samples were injected to produce a higher signal. Targeted metabolite analysis of loganic acid, loganin, secologanin, strictosidine, tabersonine, catharanthine, vindoline and vinblastine was conducted using an Advance ultra-HPLC (UHPLC) system (Bruker Daltonics) coupled to an EVOQ Elite triple quadrupole mass spectrometer (Supplementary Table 6) (Bruker Daltonics). Accurate masses of some analytes (Supplementary Methods) were analysed on a Dionex UltiMate 3000 UHPLC (Fisher Scientific) connected to an Orbitrap Fusion Mass Spectrometer (Supplementary Methods) (Thermo Fisher Scientific).

For the analysis of non-polar metabolites (geraniol and 8-hydroxygeraniol, and by-product citronellol), cells were grown in 2 ml of medium using 24-well deep plates. Geraniol and 8-hydroxygeraniol were extracted by mixing 1.6 ml samples with 400 µl of ethyl acetate (containing 50 µg ml$^{-1}$ nerol as the internal standard), shaken at 4 °C for 30 min. The aqueous and organic phases were separated by centrifugation at 13,000$g$ for 1 min, and 250 µl of the upper (ethyl acetate) phase was transferred to a gas chromatography vial with glass insert and analysed on a gas chromatography–flame ionization detector. Details for the gas chromatography and LC–MS programmes are described in the Supplementary Methods.

## Fluorescence microscopy
The yEGFP was fused to the N or C terminus of target proteins depending on their reported or predicted structural information. All yEGFP proteins were integrated into the wild-type yeast strains and confirmed with colony PCR. Localization markers for nucleus (Addgene plasmid no. 133648), endoplasmic reticulum membrane (Addgene plasmid no. 133647), mitochondria (Addgene plasmid no. 133655) and vacuole (Addgene plasmid no. 133654), containing mCherry or DuDre, were ordered from Addgene and integrated into the strains with yEGFP fusion protein as described previously[47]. Strains were cultivated in SC-Trp medium overnight (16–20 h) and imaged in the morning, or overnight culture was diluted 1:5 and cultivated for 4 h before analysis. Then 3 µl of culture was applied to an objective glass, covered with a cover glass and imaged immediately. Confocal images were acquired with a laser-scanning upright confocal microscope LSM 700, Axio Imager 2 (Carl Zeiss, Inc.), equipped with four diode lasers (405, 488, 555, 639 nm), using a C-Apochromat ×63/1.2 W Korr M27 water objective and ×4 scan zoom. Pinhole size was set to 1 Airy unit. Samples were illuminated with 488 and 555 nm lasers at 0.4–4 mW. 512 × 512 pixel images were acquired using photomultiplier tube detectors. yeGFP acquisition parameters were: 300–578 nm (emission wavelength range), 0.39 µs (line time), 600–1,000 (gain) and 0 (offset). mCherry/DuDre acquisition parameters were: 578–800 nm (emission wavelength range), 0.39 µs (line time), 600–1.000 (gain) and 0 (offset). A line average of eight was applied to both channels. Images were processed with ZEN v.3.2 (blue edition) (Carl Zeiss Microscopy GmbH).

## Analysis of proteins
All yeast strains were grown in 2 ml of SC or YPD media in three biological replicates at 30 °C 300 r.p.m. for 2 days. Cells (roughly 50 OD × ml units) were pelleted by centrifugation at 5,000$g$ for 3 min and kept at −80 °C until lysis and sample preparation. Yeast cells were lysed and protein extracted as described previously[48]. Briefly, it consists of a bead-beating step, cell lysis, protein precipitation, protein resuspension, protein quantification and normalization of protein concentration followed by standard bottom-up proteomic procedures of reducing and blocking cysteine residues and tryptic digestion. Peptide samples were analysed on an Agilent 1290 UHPLC system coupled to an Agilent 6460QQQ MS (Agilent Technologies) as described previously[49]. Briefly, peptide samples were loaded onto an Ascentis ES-C18 Column (Sigma–Aldrich) and introduced to the MS using a Jet Stream source (Agilent Technologies) operating in positive-ion mode. The data were acquired with Agilent MassHunter Workstation Software v.B.08.02. LC–MS Data Acquisition operating in dynamicMRM mode. Multiple reaction monitoring (MRM) transitions for the targeted proteins were generated by Skyline software v.20.2 (MacCoss Laboratory Software) and selection criteria excluded peptides with Met/Cys residues, tryptic peptides followed by further cut sites (KK/RR) and peptides with proline adjacent to K/R cut sites. The data and Skyline methods are available on Panoramaweb (https://panoramaweb.org/microbial-synthesis-of-vinblastine.url). Proteins from the MIA-DJ (tabersonine-vinblastine double module) strain were analysed using the Cap-LC system equipped with a C18 easy spray column (Thermo Fisher Scientific), coupled to Orbitrap Q Exactive HF-X mass spectrometer (Thermo Fisher Scientific)[50]. The resulting data were analysed using Proteome Discover v.2.3 (Thermo Fisher Scientific) by searching against the *S. cerevisiae* proteome data (Uniprot ID UP000002311) combined with all heterologous protein sequences. The abundance of each protein is reported as the relative intensity to the total intensity of all identified peptides.

## Reporting summary

Further information on research design is available in the Nature Research Reporting Summary linked to this article.

## Data availability

All metabolite data shown in figures and extended data figures are available in the Source data provided with this paper. Accession of all heterologous genes used in this study are listed in Supplementary Table 1. Targeted proteomics data for strain MIA-AU are available at ProteomeXchange with identifier PXD024976 (https://panoramaweb.org/microbial-synthesis-of-vinblastine.url). Untargeted proteomics data for strain MIA-DJ are available at ProteomeXchange with identifier PXD025067.

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

**Acknowledgements** This study was supported by grants from the Novo Nordisk Foundation (grant no. NNF10CC1016517), Horizon 2020 (MIAMi, grant no. 814645) and BII Foundation (grant nos. NNF19OC0055591 and NNF20SA0067054). We thank S. Rose (Axyntis Group, France) for providing the tabersonine standard. We thank P. Charusanti and L. Ding for technical assistance. Confocal microscopy analysis was carried out at Center for Advanced Bioimaging, University of Copenhagen.

**Author contributions** J.Z., M.K.J. and J.D.K. conceived the project and wrote the manuscript. J.Z., L.G.H., O.G. and K.V. designed the experiments. J.Z., L.G.H., O.G., K.V., L.M.M.L., L.S., K.B.A., P.R., E.C.-A., L.C., V.D., B.L., A.K.H., S.N., K.G., M.L., D.A., M.A.K.J., L.J.G.C., M.K., H.B.C., S.S., E.B. and T.W. performed experiments. J.Z., L.G.H., O.G., K.V., L.S., K.B.A., P.R., E.C.-A., L.C., S.N., M.K., H.B.C., E.B., C.J.P., M.K.J. and T.W. analysed the results. L.G.H., K.V., S.E.O., E.A.S. and V.C. identified biosynthetic gene sequences.

**Competing interests** J.D.K., J.Z., L.G.H., K.V., V.D., S.E.O. and M.K.J. are inventors on pending patent applications. J.D.K. has a financial interest in Amyris, Lygos, Demetrix, Napigen, Apertor Pharmaceuticals, Maple Bio, Ansa Biotechnologies, Berkeley Yeast and Zero Acre Farms. The other authors declare no competing interests.

**Additional information**
**Correspondence and requests for materials** should be addressed to Michael K. Jensen or Jay D. Keasling.

**a**

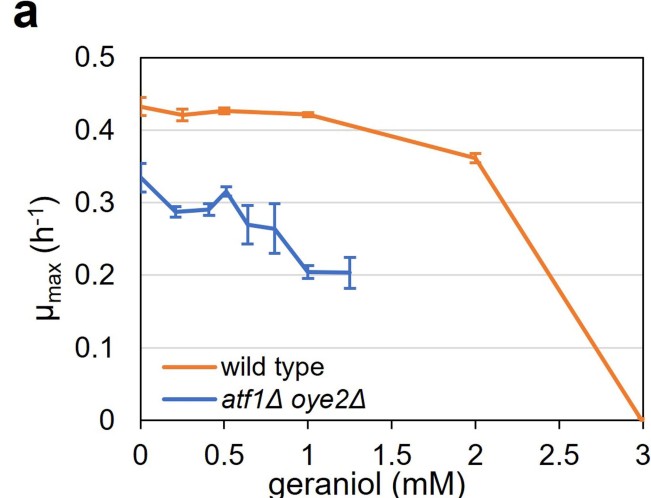
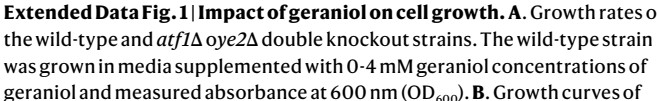

**b**

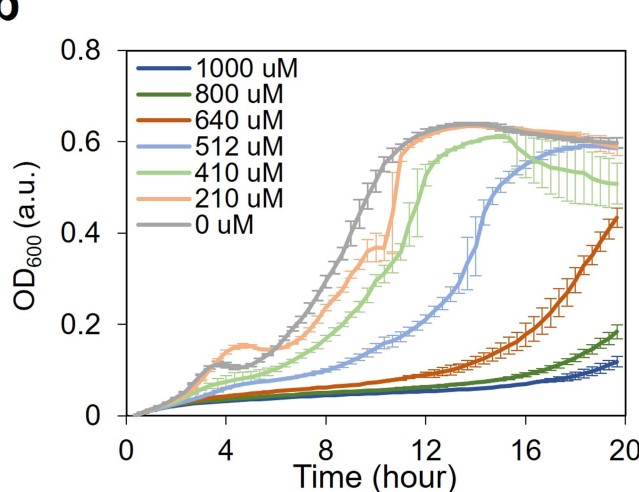

**Extended Data Fig. 1 | Impact of geraniol on cell growth. A**. Growth rates of the wild-type and *atf1Δ oye2Δ* double knockout strains. The wild-type strain was grown in media supplemented with 0-4 mM geraniol concentrations of geraniol and measured absorbance at 600 nm (OD$_{600}$). **B**. Growth curves of *atf1Δ oye2Δ* strain in the presence of 0-1 mM geraniol. Higher concentrations of geraniol caused retarded growth and a dramatically longer lag-phase. Data presented are mean ± s.d. ($n = 3$).

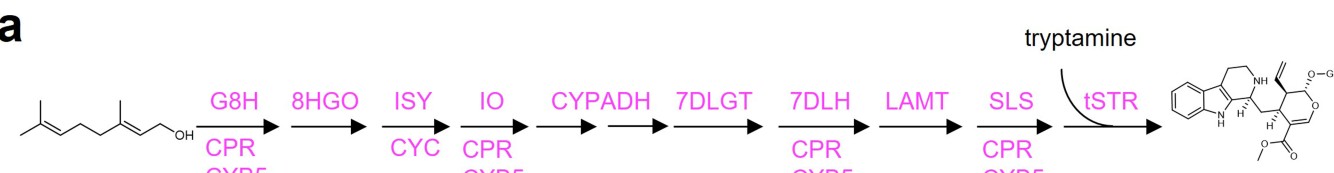

**a**

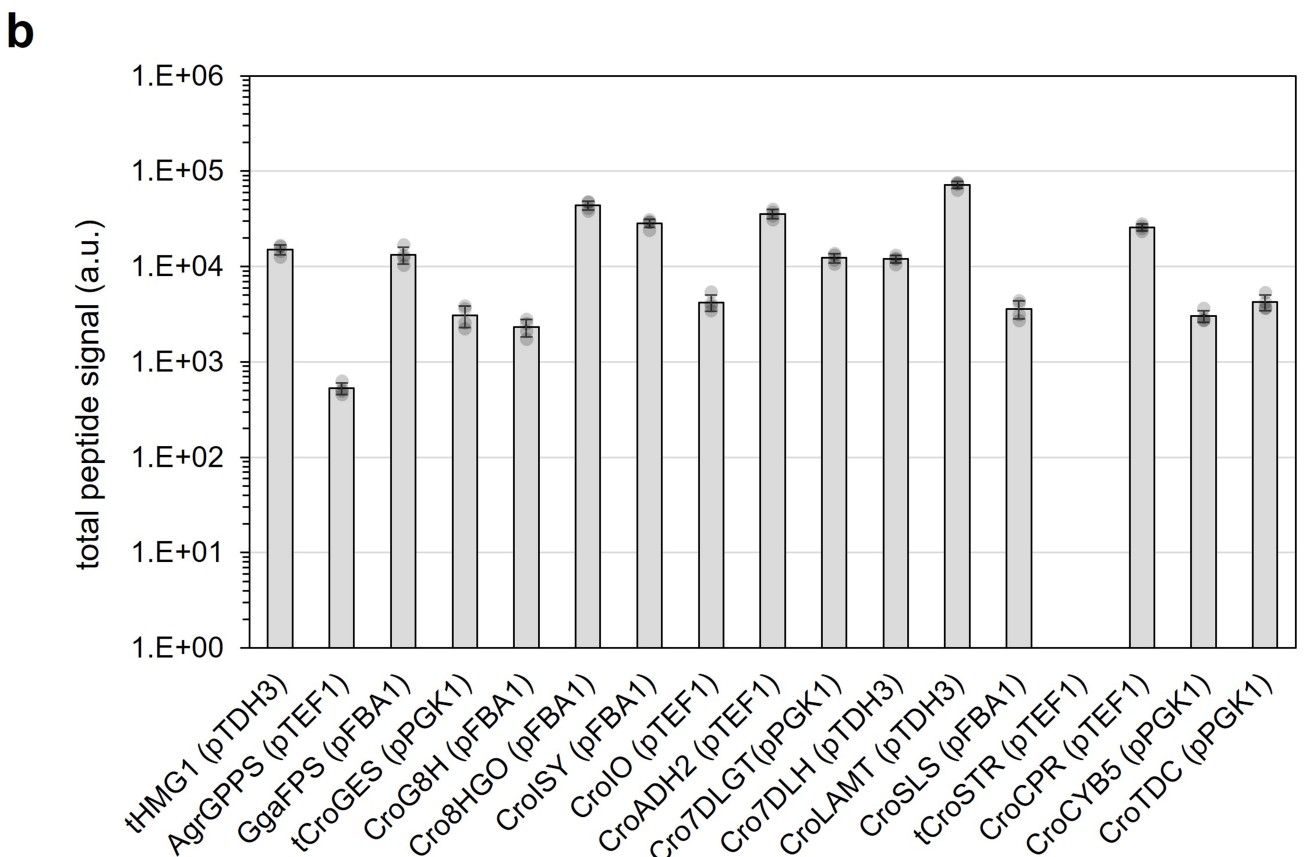

**b**

**Extended Data Fig. 2 | Analysis of proteins in the strictosidine module. a**. All biosynthetic genes in the strictosidine module were integrated into the wild-type strain CEN.PK2-1C; **b**. Measurement of protein abundance in strain MIA-AU using a shotgun targeted proteomics approach. *Cro*STR was below the detection limit. Data presented are mean ± s.d. (*n* = 3-4).

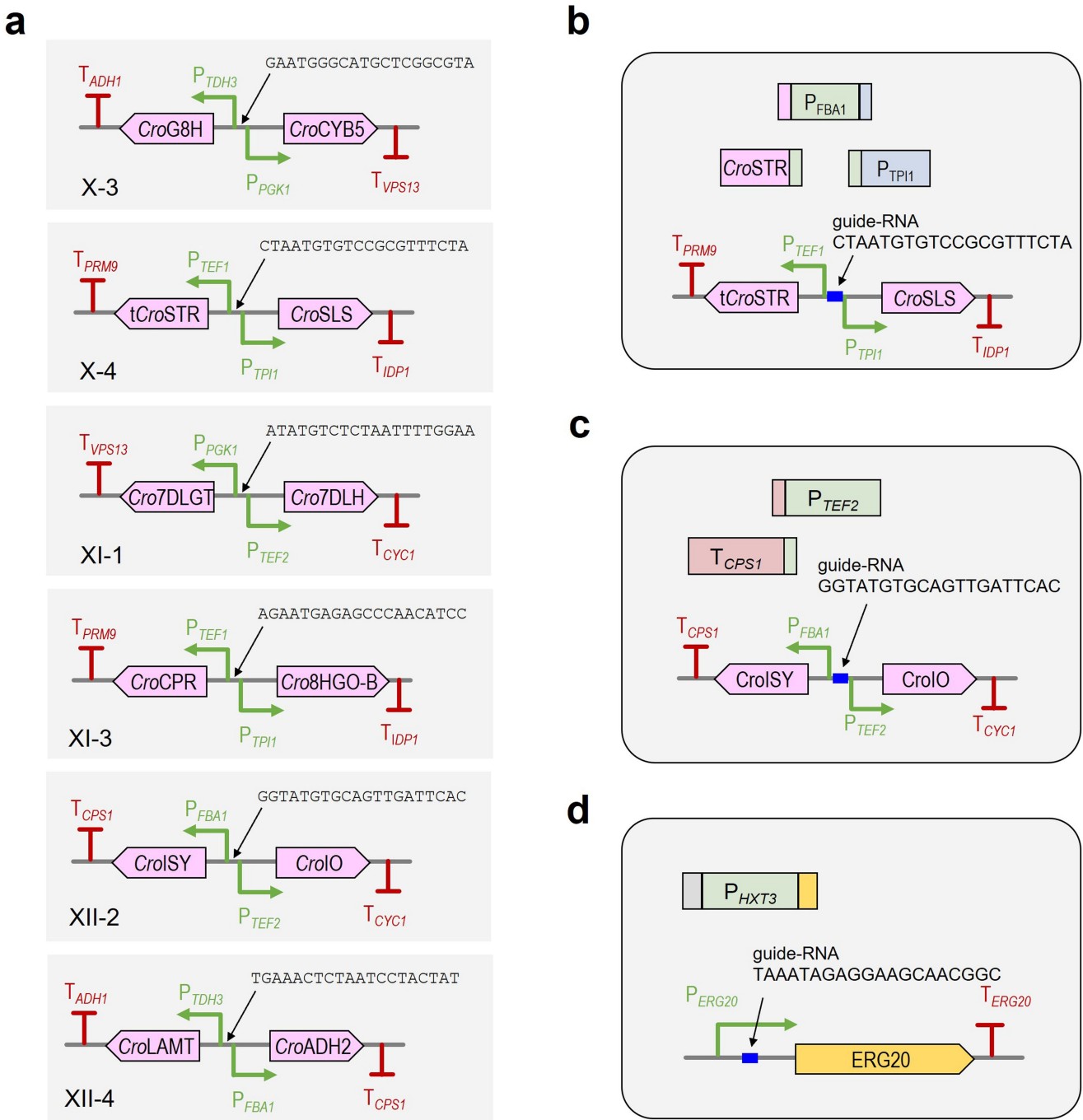

**Extended Data Fig. 3 | Design of EZ-Swap strains. a**. Genetic design of EZ-Swap base strain MIA-BG; **b**. Strategy for swapping t*Cro*STR with full length *Cro*STR; **c**. Strategy for knocking out *Cro*ISY; **d**. Strategy for replacing P$_{ERG20}$ with P$_{HXT3}$ for dynamic knockdown of *ERG20*.

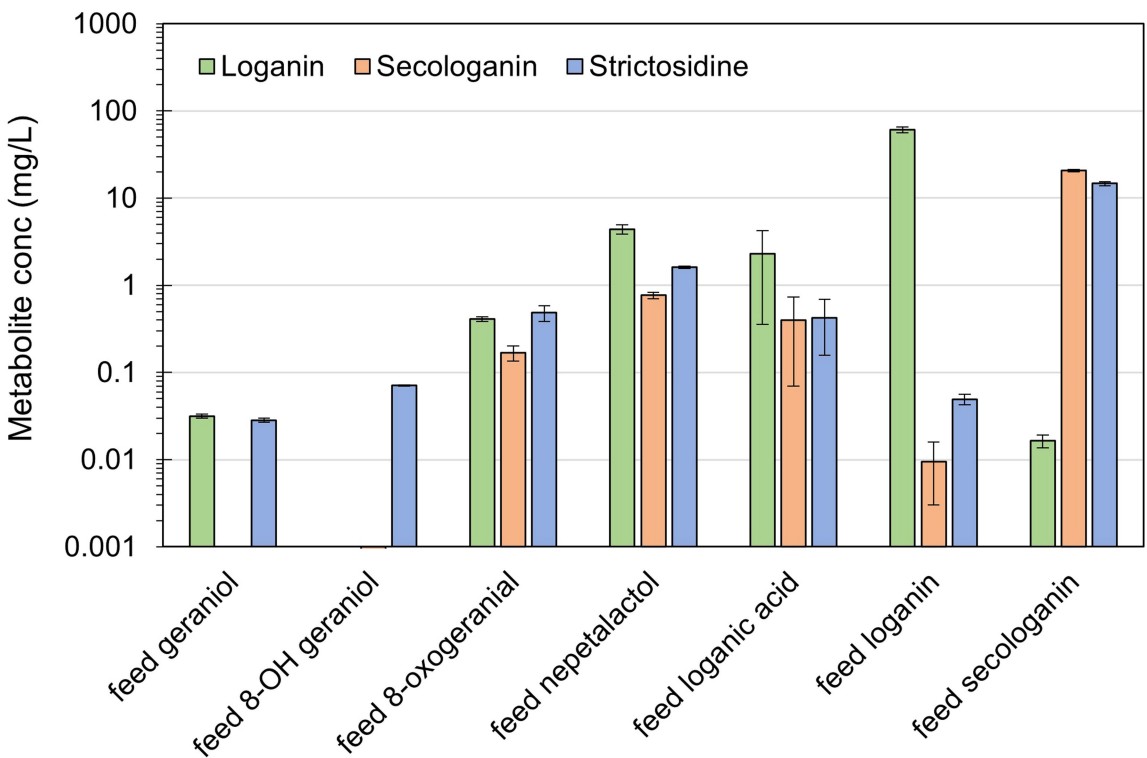

**Extended Data Fig. 4 | Strictosidine production in MIA-BG strain by feeding tryptamine and one of the strictosidine precursors.** Data presented are mean ± s.d. (n = 3).

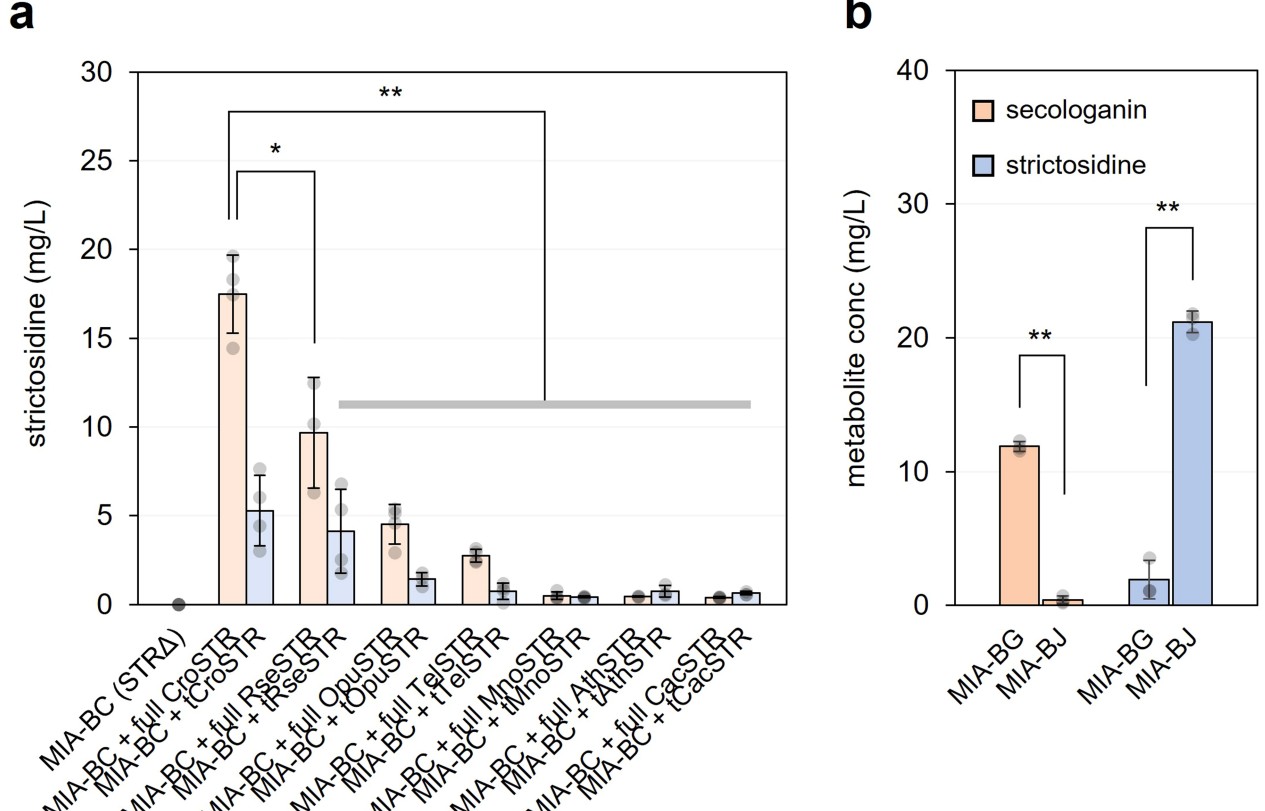

**Extended Data Fig. 5 | Screening of strictosidine synthase homologues.**
**a.** Screening full-length and truncated STR variants in a strain without STR (MIA-BC); **b.** Conversion of secologanin to strictosidine by truncated *Cro*STR and full-length *Cro*STR. Data presented are mean ± s.d. ($n = 3$). * p-value < 0.05; ** p-value < 0.001. Student's two-tailed t-test. More statistical analysis is available in the source data file.

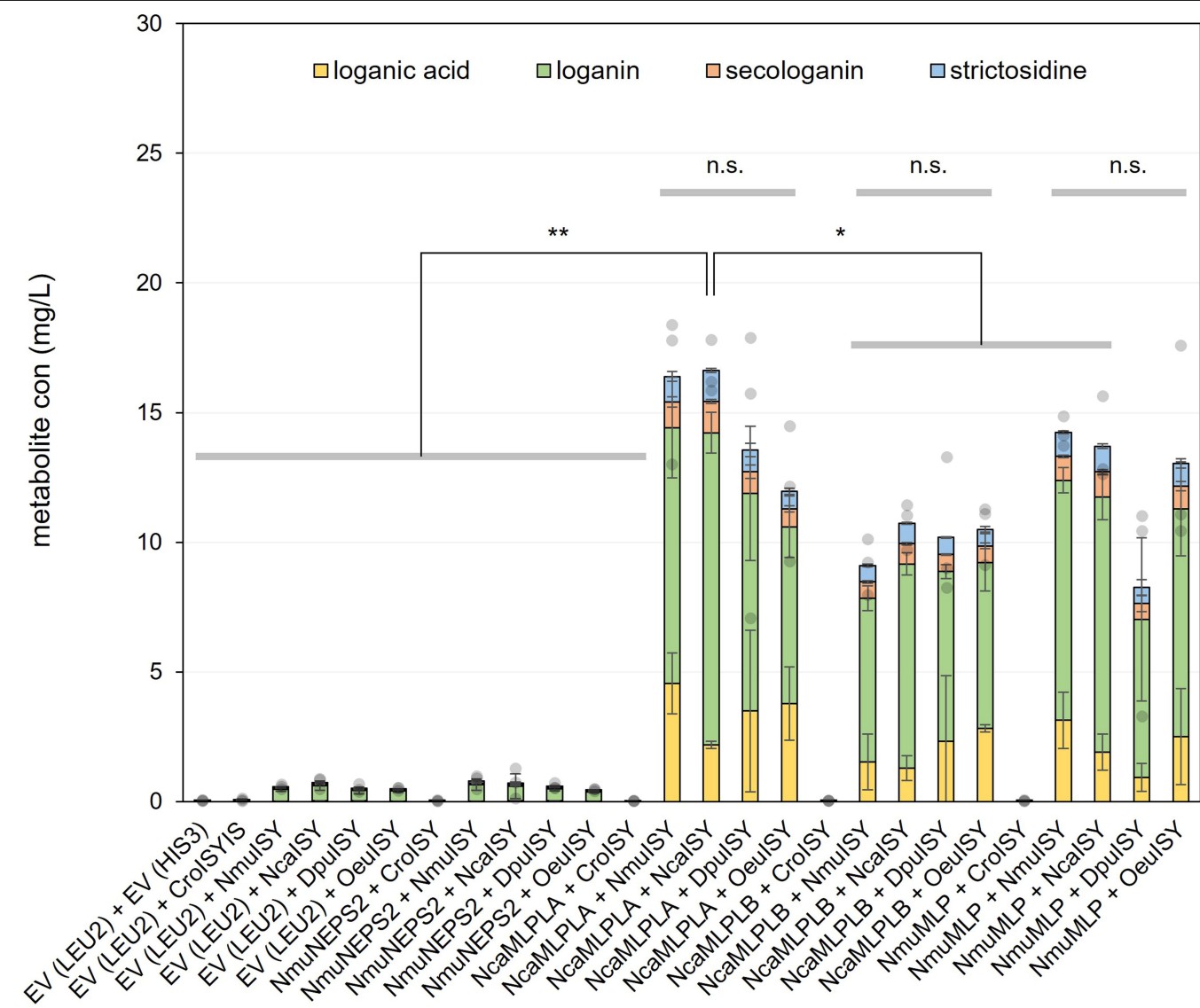

**Extended Data Fig. 6 | Screening of iridoid synthase (ISY) and cyclase (CYC) combinations for efficient conversion of 8-oxogeranial to *cis-trans*-nepetalactol.** Combinations of ISY and CYC were transformed into the MIA-BKV-1 strain (MIA-BJ without ISY/CYC). All ISYs and CYCs are under the control of *GAL1* or *GAL10* promoters. All yeast transformants (three biological replicates) were grown in SC medium (lacking histidine and leucine for selection of the two plasmids) containing 1.5% glucose and 1.5% galactose as carbon sources. All cultures were supplemented with 100 μM 8-oxogeranial and 100 μM tryptamine. After 3 days, additional 0.5% galactose and 100 μM 8-oxogeranial were added to the culture. Loganic acid, loganin, secologanin and strictosidine were measured and combined values were used as an approximate for total production of *cis-trans*-nepetalactol. EV, empty vector. Data presented are mean ± s.d. ($n = 3$). n.s. not significant; * $p$-value < 0.05; ** $p$-value < 0.01. Student's two-tailed t-test. More statistical analysis is available in the source data file.

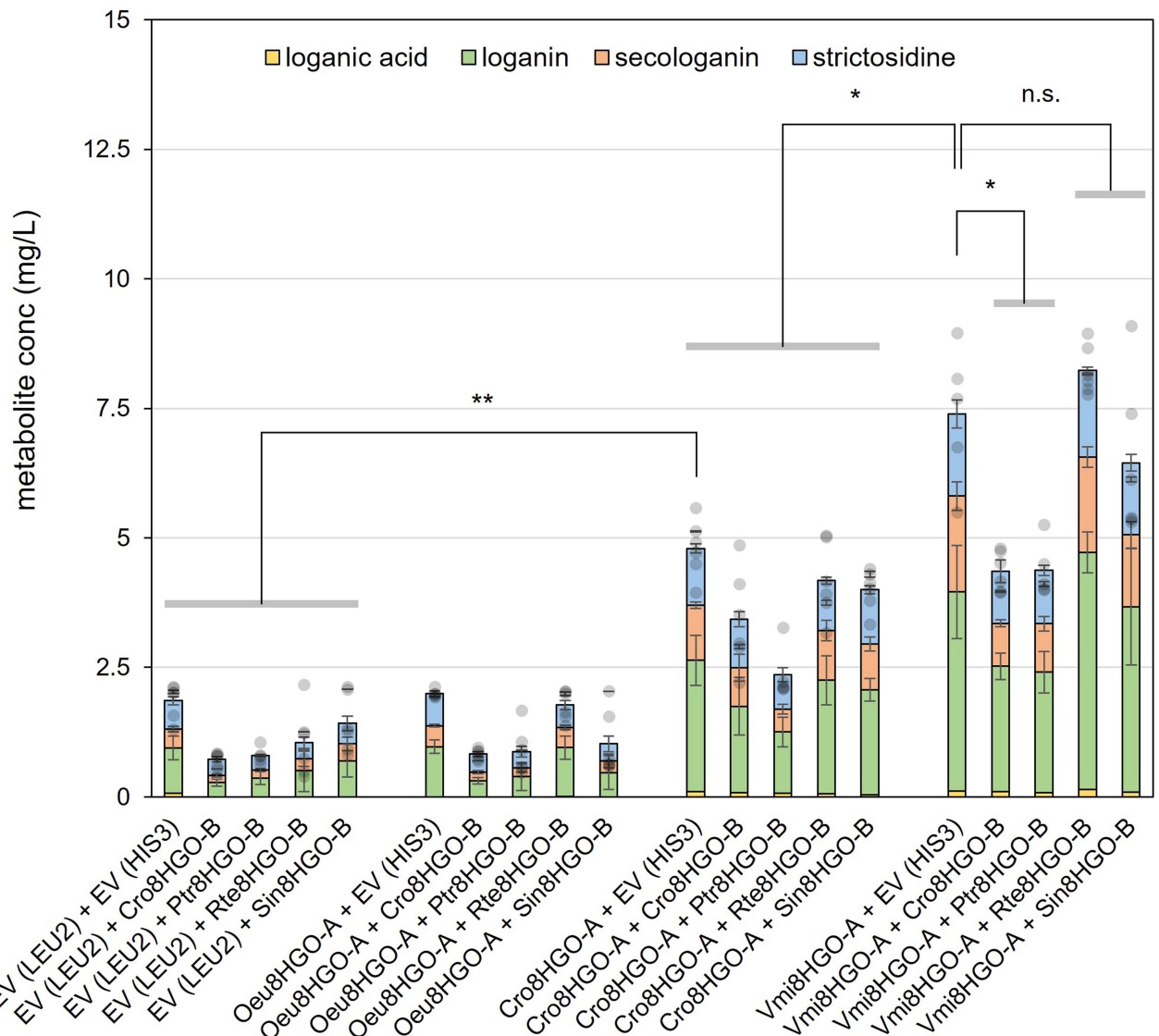

**Extended Data Fig. 7 | Screening of 8HGO variants for efficient conversion of 8-hydroxygeraniol to 8-oxogeranial.** A combinatorial library of homologues of two 8HGO variants (A-type: NAD+-dependent, B-type: NADP+-dependent) were transformed into MIA-BKV-5 strain (8HGO deleted from strain MIA-BS-5). All 8HGO variants are under the control of *GAL1* or *GAL10* promoters. All yeast transformants were grown in SC medium (lacking histidine and leucine for selection of the two plasmids) containing 1% glucose and 1% galactose as carbon sources. All cultures were supplemented with 100 μM 8-hydroxygeraniol and 1 mM tryptamine and grown for 6 days. Loganic acid, loganin, secologanin and strictosidine were measured and combined as an indicator for total production of 8-hydroxygeraniol. EV, empty vector. Data presented are mean ± s.d. for each metabolite (n = 5-6). n.s. not significant; * *p*-value < 0.05; ** *p*-value < 0.01. Student's two-tailed t-test. More statistical analysis is available in the source data file.

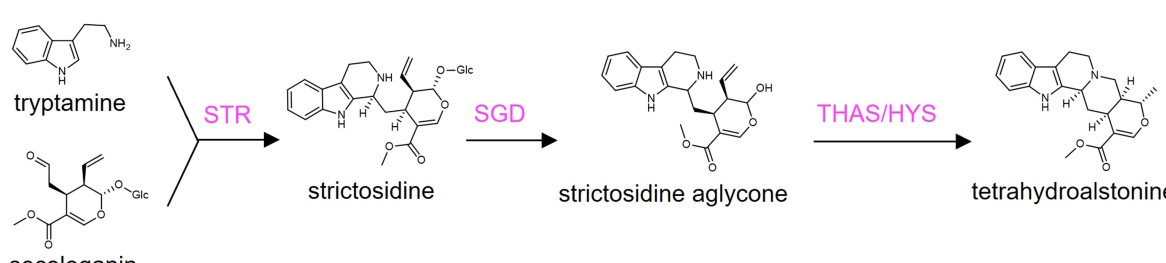

**a**

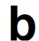

**b**

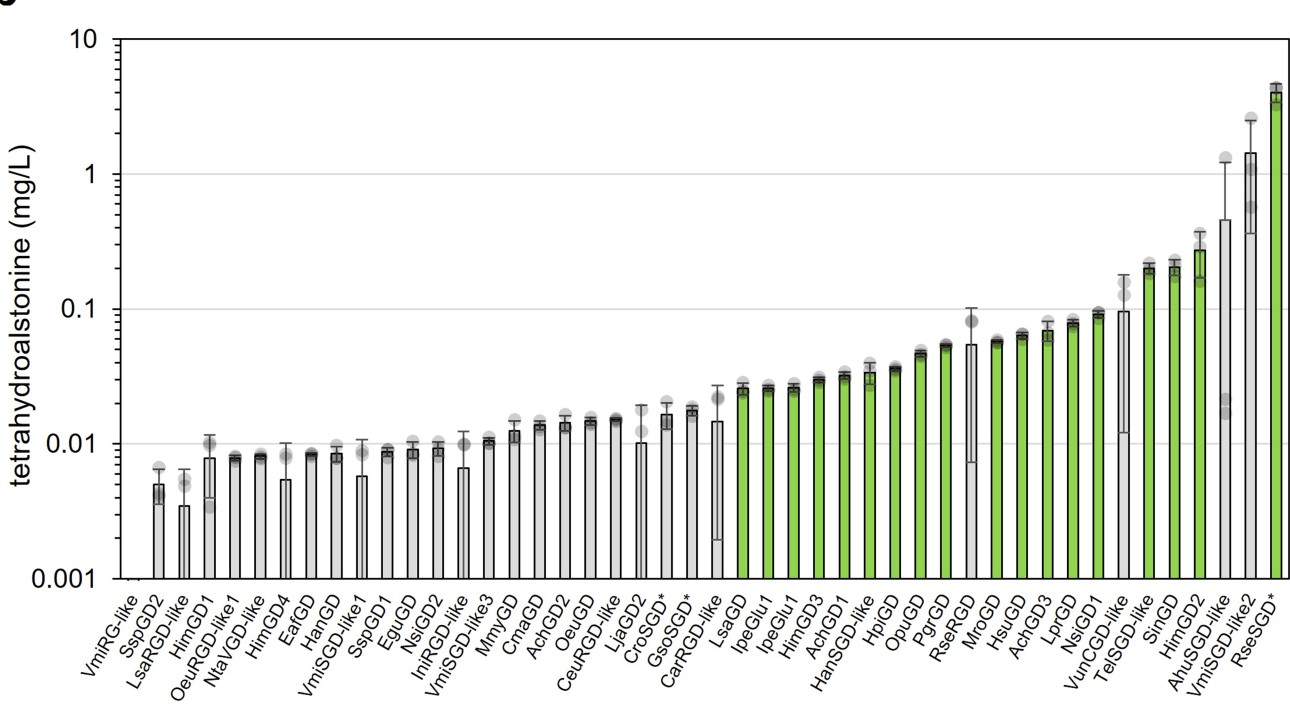

**Extended Data Fig. 8 | Screening of SGD homologs. a**. tetrahydroalstonine biosynthesis pathway from tryptamine and secologanin; **b**. Production of THA in yeast strains expressing *Cro*STR, *Cro*HYS and one of the SGD homologs (Supplementary Table 1). Asterisk indicates the data were from a different experiment but under the same condition (SC medium + 100 μM secologanin + 1 mM tryptamine, cultivated for six days and analyzed on LC-MS). Data presented are mean ± s.d. (n = 3). Green color indicates significantly higher production compared to CroSGD (*p*-value < 0.05, Student's two-tailed t-test). More statistical analysis is available in the source data file.

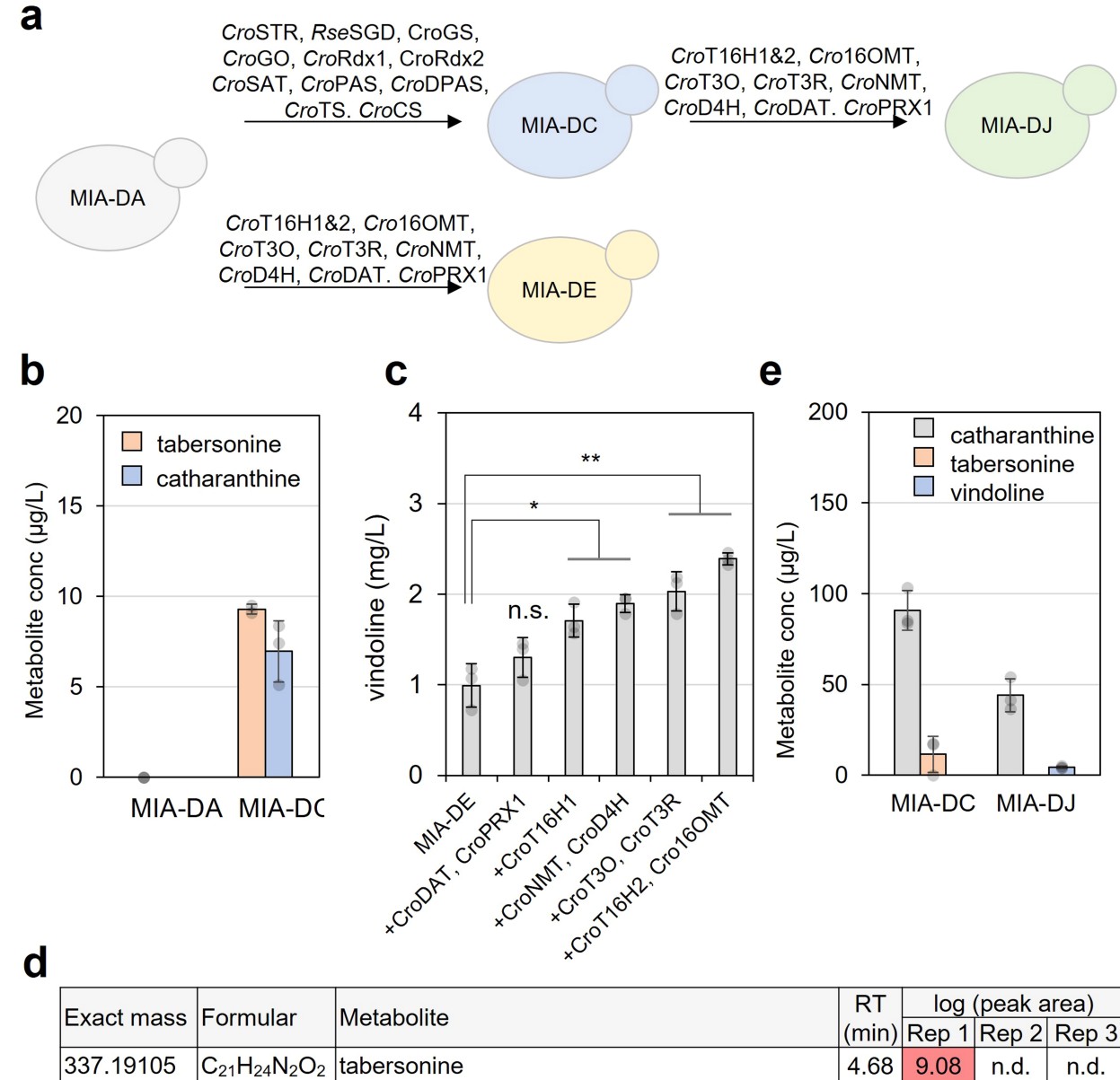

| Exact mass | Formular | Metabolite | RT (min) | log (peak area) Rep 1 | Rep 2 | Rep 3 |
|---|---|---|---|---|---|---|
| 337.19105 | $C_{21}H_{24}N_2O_2$ | tabersonine | 4.68 | 9.08 | n.d. | n.d. |
| 353.18597 | $C_{21}H_{24}N_2O_3$ | 16-hydroxytabersonine | 4.08 | 9.55 | 9.67 | 9.71 |
| 367.20162 | $C_{22}H_{26}N_2O_3$ | 16-methoxytabersonine | 4.73 | 9.55 | 9.49 | 9.49 |
| 385.21218 | $C_{21}H_{26}N_2O_3$ | 3-hydroxy-16-methoxy-2,3-dihydrotabersonine | 4.54 | 8.79 | 7.99 | 8.03 |
| 399.22783 | $C_{23}H_{30}N_2O_4$ | deacetoxyvidoline | 4.81 | n.d. | 9.04 | 9.04 |
| 415.22275 | $C_{23}H_{30}N_2O_5$ | deacetylvinddoline | 4.36 | n.d. | 6.79 | 6.79 |
| 457.23331 | $C_{25}H_{32}N_2O_6$ | vindoline | 4.75 | n.d. | 8.45 | 8.43 |

**Extended Data Fig. 9 | Functional expression of the tabersonine/ catharanthine and vindoline modules. a.** Construction of yeast strains for the functional expression of the tabersonine/catharanthine module (MIA-DC), the vindoline module (MIA-DE), and the tabersonine/catharanthine+vindoline modules (MIA-DJ); **b.** Production of catharanthine and tabersonine from fed 0.05 mM secologanin and 1 mM tryptamine; **c.** Production of vindoline from fed 0.05 mM tabersonine by strain MIA-DE expressing an additional copy of one or two genes from a centromeric plasmid; d. production of catharanthine, tabersonine and vindoline in MIA-DC and MIA-DJ strains fed with 0.1 mM secologanin and 1 mM tryptamine. e. Peak area of vindoline module intermediates in the cultivation of strain MIA-DE fed with 0.1 mM catharanthine and tabersonine. Data presented are mean ± s.d. (n = 3). * $p$-value < 0.05; ** $p$-value < 0.01; n.s., not significant. More statistical analysis is available in the source data file.

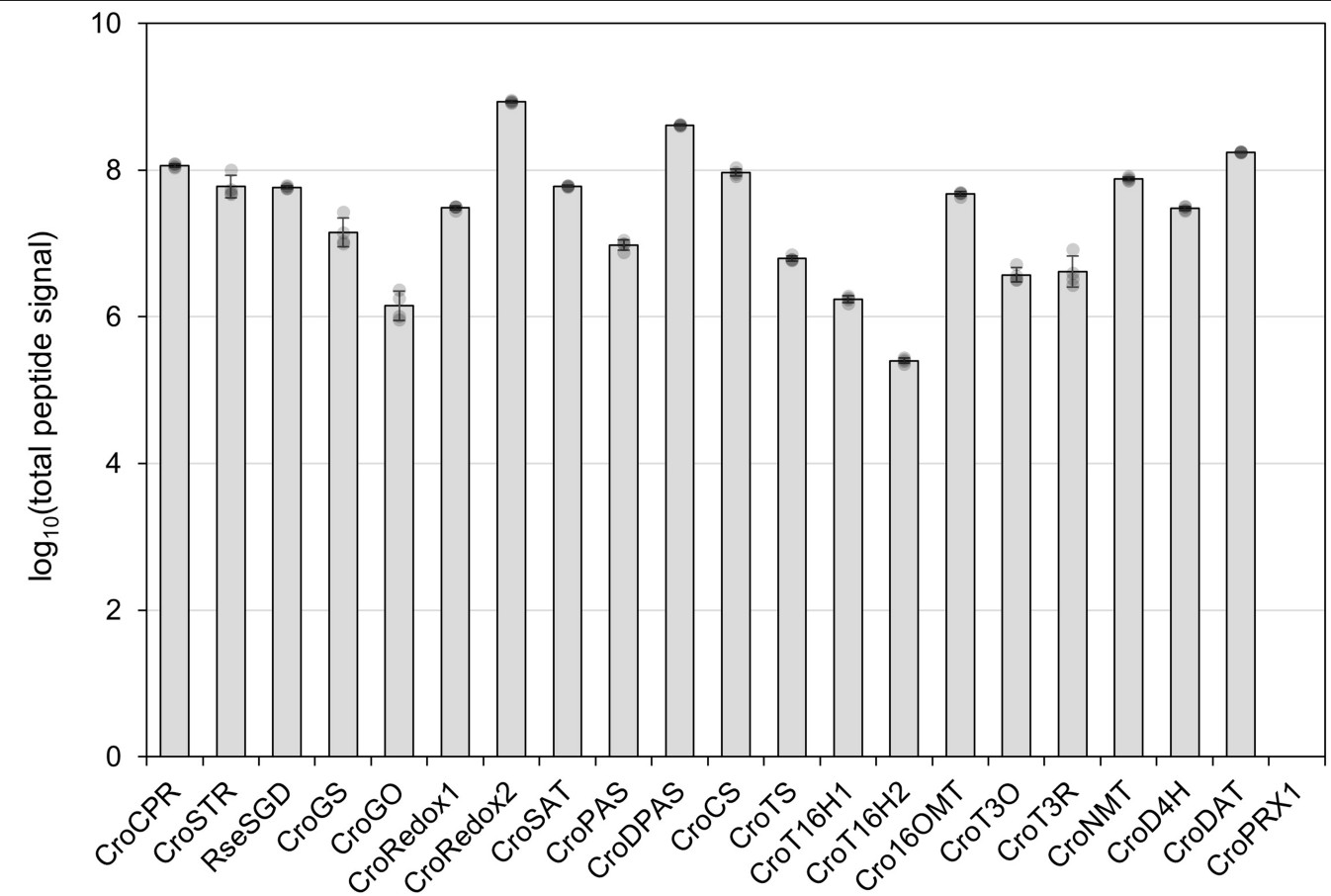

**Extended Data Fig. 10 | Relative abundance of all proteins involved in the tabersonine and vindoline modules.** The average value of unique peptide sequences for each protein was determined by untargeted proteomics from four biological replicates. All protein abundances were baseline corrected using background signals detected from a wild-type strain without any MIA genes. *Cro*PRX1 was not detected in any of the replicates. Data presented are mean ± s.d. (*n* = 3).

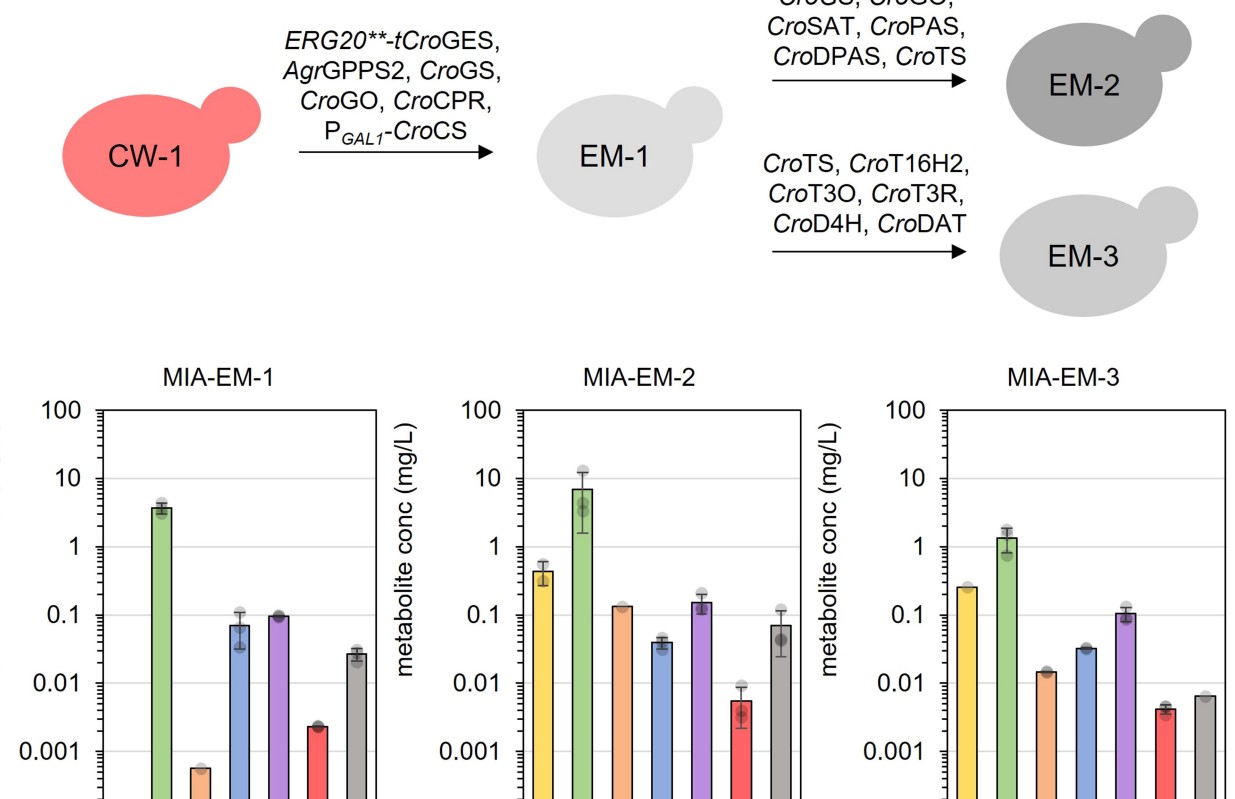

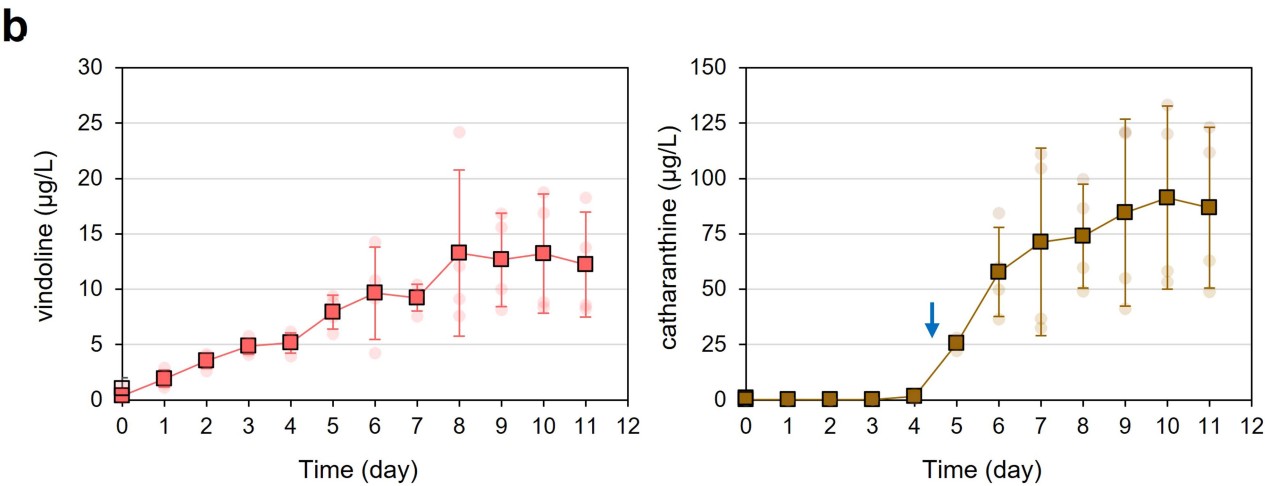

**Extended Data Fig. 11 | *De novo* production of catharanthine and vindoline in fed-batch cultivations. a**. *De novo* vindoline production by MIA-CW-1 and MIA-EM strain series in BioLector Pro (1 mL) fed-batch cultivations. **b**. *De novo* vindoline and catharanthine production by fed-batch cultivations of strain MIA-EM-2 in ambr®250 bioreactors. The blue arrow in the catharanthine panel indicates the start of galactose induction. Data presented are all replicates (dots), and mean (square box) ± s.d. (n = 4).

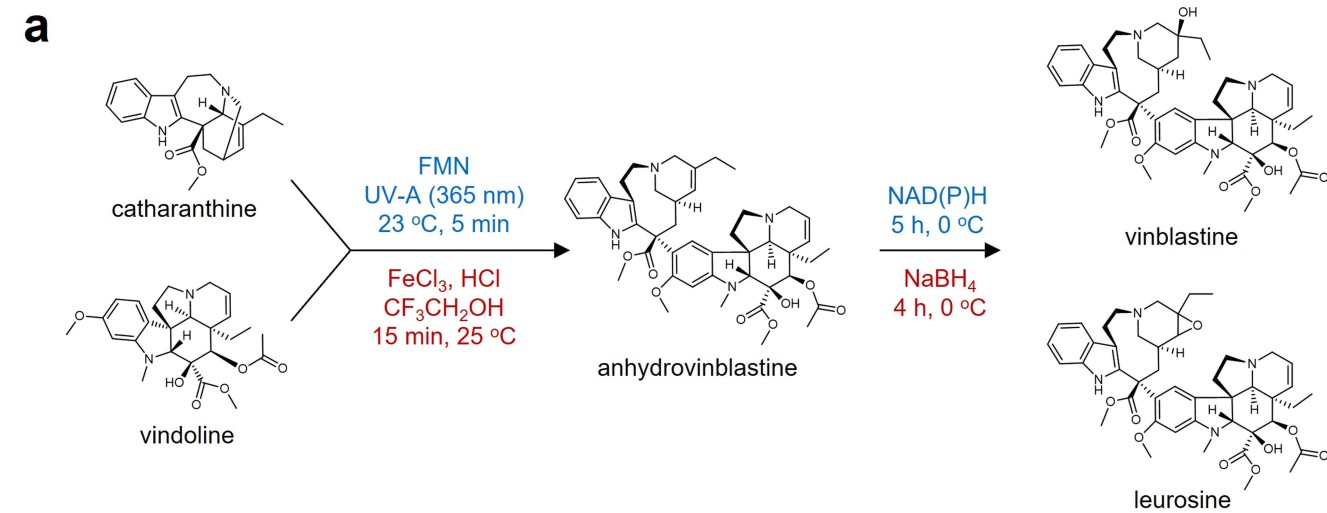

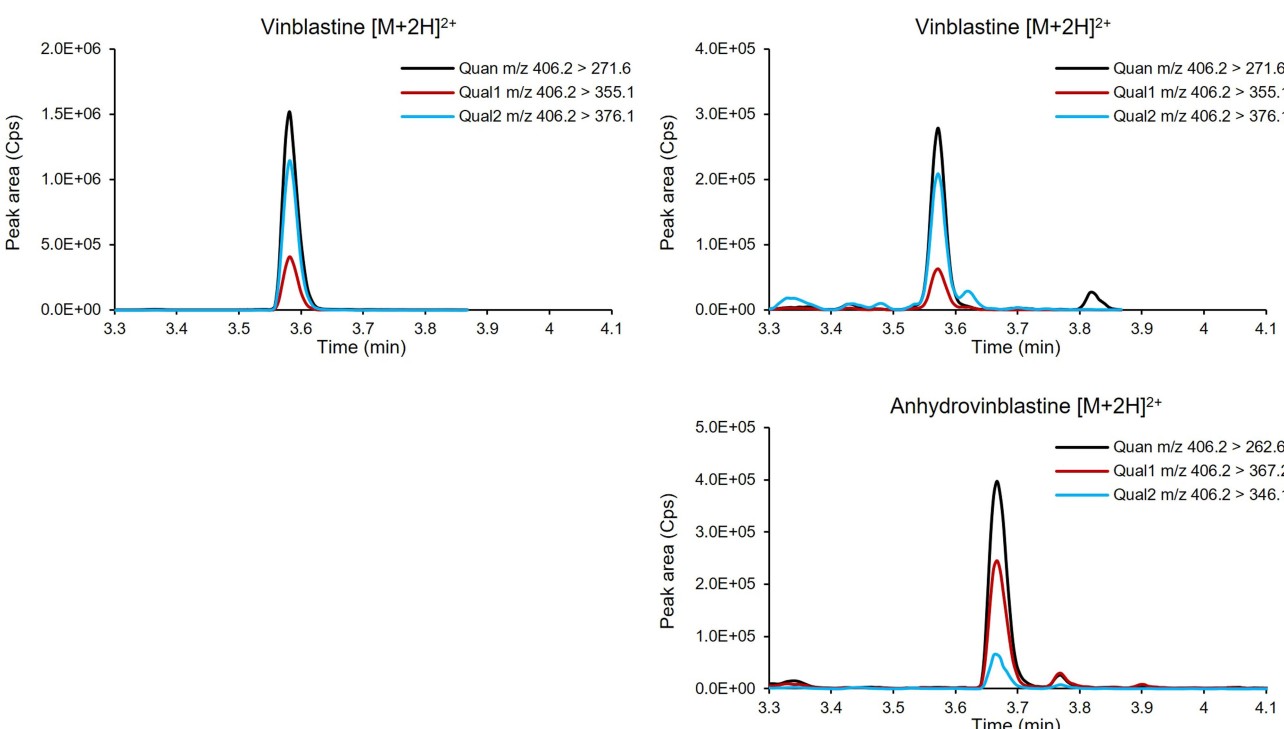

**Extended Data Fig. 12 | Semi-synthesis of vinblastine by coupling of vindoline and catharanthine. a**. Semi-synthesis of vinblastine by chemical coupling of catharanthine and vindoline. Two methods were used for the coupling of vindoline and catharanthine, a photo-chemical coupling under near-UV light[38] followed by reduction using NADH, and a chemical coupling using FeCl$_3$, followed by reduction using NaBH$_4$[2]; **b**. Representative chromatograms of vinblastine standard (left), and vinblastine and anhydrovinblastine produced from the Fe(III)-based chemical coupling using yeast-produced catharanthine and vindoline (right). FMN: flavin mononucleotide.

# Reporting Summary

## Statistics

For all statistical analyses, confirm that the following items are present in the figure legend, table legend, main text, or Methods section.

| n/a | Confirmed | |
|---|---|---|
| ☐ | ☒ | The exact sample size (*n*) for each experimental group/condition, given as a discrete number and unit of measurement |
| ☐ | ☒ | A statement on whether measurements were taken from distinct samples or whether the same sample was measured repeatedly |
| ☐ | ☒ | The statistical test(s) used AND whether they are one- or two-sided<br>*Only common tests should be described solely by name; describe more complex techniques in the Methods section.* |
| ☒ | ☐ | A description of all covariates tested |
| ☒ | ☐ | A description of any assumptions or corrections, such as tests of normality and adjustment for multiple comparisons |
| ☐ | ☒ | A full description of the statistical parameters including central tendency (e.g. means) or other basic estimates (e.g. regression coefficient) AND variation (e.g. standard deviation) or associated estimates of uncertainty (e.g. confidence intervals) |
| ☒ | ☐ | For null hypothesis testing, the test statistic (e.g. *F*, *t*, *r*) with confidence intervals, effect sizes, degrees of freedom and *P* value noted<br>*Give P values as exact values whenever suitable.* |
| ☒ | ☐ | For Bayesian analysis, information on the choice of priors and Markov chain Monte Carlo settings |
| ☒ | ☐ | For hierarchical and complex designs, identification of the appropriate level for tests and full reporting of outcomes |
| ☒ | ☐ | Estimates of effect sizes (e.g. Cohen's *d*, Pearson's *r*), indicating how they were calculated |

*Our web collection on statistics for biologists contains articles on many of the points above.*

## Software and code

Policy information about availability of computer code

| | |
|---|---|
| Data collection | The targeted proteomics data were acquired with Agilent MassHunter Workstation Software version B.08.2, LC/MS Data Acquisition operating in dynamic MRM mode. MRM transitions for the targeted proteins were generated by Skyline software 20.2 (MacCoss Lab Software) and selection criteria excluded peptides with Met/Cys residues, tryptic peptides followed by additional cut sites (KK/RR), and peptides with praline adjacent to K/R cut sites. The data and Skyline methods are available on Panoramaweb (https://panoramaweb.org/microbial-synthesis-of-vinblastine. url).Proteins from the MIA-DJ (tabersonine-vinblastine double module) strain were analyzed using Cap-LC system equipped with a C18 easy spray column (Thermo Fisher Scientific, MA), coupled to Orbitrap Q Exactive HF-X mass spectrometer (Thermo Fisher Scientific) (Wright et al. 2020). |
| Data analysis | The untargeted proteomics data from MIA-DJ strain were analyzed using Proteome Discover 2.3 (Thermo Fisher Scientific) by searching against the S. cerevisiae proteome data (Uniprot ID UP000002311) combined with all heterologous protein sequences. The abundance of each protein is reported as the relative intensity with respect to total intensity of all identified peptides. For GC-FID data analysis Chromeleon version 7.2.1 was used, while LC-MS data was analysed using the MS Workstation 8.2.1 software. |

For manuscripts utilizing custom algorithms or software that are central to the research but not yet described in published literature, software must be made available to editors and reviewers. We strongly encourage code deposition in a community repository (e.g. GitHub). See the Nature Portfolio guidelines for submitting code & software for further information.

## Data

Policy information about availability of data

All manuscripts must include a data availability statement. This statement should provide the following information, where applicable:
- Accession codes, unique identifiers, or web links for publicly available datasets
- A description of any restrictions on data availability
- For clinical datasets or third party data, please ensure that the statement adheres to our policy

All metabolite data shown in figures and extended data figures are available as Source Data. Accession of all heterologous genes used in this study are listed in Supplementary Table 1. Targeted proteomics data for strain MIA-AU are available at ProteomeXchange with identifier PXD024976 (https://panoramaweb.org/Panorama%20Public/2021/JBEI%20-%20J.%20Zhang%20et%20al.%20-%20vinblastine%20paper/project-begin.view?). Untargeted proteomics data for strain MIA-DJ are available at ProteomeXchange with identifier PXD025067.

## Human research participants

Policy information about studies involving human research participants and Sex and Gender in Research.

| | |
|---|---|
| Reporting on sex and gender | N/A |
| Population characteristics | N/A |
| Recruitment | N/A |
| Ethics oversight | N/A |

Note that full information on the approval of the study protocol must also be provided in the manuscript.

# Field-specific reporting

Please select the one below that is the best fit for your research. If you are not sure, read the appropriate sections before making your selection.

☒ Life sciences          ☐ Behavioural & social sciences          ☐ Ecological, evolutionary & environmental sciences

For a reference copy of the document with all sections, see nature.com/documents/nr-reporting-summary-flat.pdf

# Life sciences study design

All studies must disclose on these points even when the disclosure is negative.

| | |
|---|---|
| Sample size | All metabolites and proteins presented in Figures and Extended Data Figures were from 3-6 biological replicates (i.e., independent culture from individual colonies). Based on standard deviation among replicates we regard this number of biological replicates to represent a valid trade-off between approximating the true mean and instrumentation band-width. |
| Data exclusions | No data were excluded from analysis. All data used can be found in Source data. |
| Replication | All measurements of metabolites and proteins were performed with 3-6 biological replicates (i.e., independent culture from individual colonies). All attempts of replication were included in the manuscript, and as such successfully used to draw conclusions based on statistical testing. |
| Randomization | Randomization was applied by randomly selecting colonies. |
| Blinding | Beyond randomization of colony picking, no blinding was applied as the data generated from random colony picking is expected to approach or even represent population means. |

# Reporting for specific materials, systems and methods

We require information from authors about some types of materials, experimental systems and methods used in many studies. Here, indicate whether each material, system or method listed is relevant to your study. If you are not sure if a list item applies to your research, read the appropriate section before selecting a response.

## Materials & experimental systems

| n/a | Involved in the study |
|-----|----------------------|
| ☒ ☐ | Antibodies |
| ☐ ☒ | Eukaryotic cell lines |
| ☒ ☐ | Palaeontology and archaeology |
| ☒ ☐ | Animals and other organisms |
| ☒ ☐ | Clinical data |
| ☒ ☐ | Dual use research of concern |

## Methods

| n/a | Involved in the study |
|-----|----------------------|
| ☒ ☐ | ChIP-seq |
| ☒ ☐ | Flow cytometry |
| ☒ ☐ | MRI-based neuroimaging |

## Eukaryotic cell lines

Policy information about cell lines and Sex and Gender in Research

| | |
|---|---|
| Cell line source(s) | All strains used in this study were derived from the wild type yeast (Saccharomyces cerevisiae) strain CEN.PK2-1C (EuroSCARF3 0000A). |
| Authentication | All yeast strains with chromosomal editing were validated by genotyping PCR and sequencing of the modified loci. |
| Mycoplasma contamination | Not applicable. |
| Commonly misidentified lines (See ICLAC register) | Not applicable. |

