## [Peer Review File · Nature]

Manuscript Title: A microbial supply chain for production of anti-cancer drug vinblastine

Reviewer Comments & Author Rebuttals

Reviewer Reports on the Initial Version:

Referee #1 (Remarks to the Author):

Monoterpene indole alkaloids (MIAs) are a diverse family of natural products featuring many compounds of global medicinal importance. In this manuscript, Zhang et al. describe the engineering of a complex, ~30-step MIA biosynthetic pathway in yeast for conversion of sugars and amino acids to vindoline and catharanthine, the two immediate precursors of the anticancer drug vinblastine. Building on earlier yeast engineering work by Brown et al. (2015), the authors first develop and optimize a strain for production of the key intermediate strictosidine using feeding experiments to identify bottlenecks, promoter tuning, combinatorial screening of ortholog libraries, and overexpression of the native geraniol pathway. The authors next used a combination of ortholog library screening, microscopy, and protein domain-swapping to identify a glucosidase enzyme that retains its function in yeast. Finally, the authors implemented a parallel pathway construction strategy to separately assemble the ~10 and ~20 genes required for de novo biosynthesis of catharanthine and vindoline, respectively, into their strictosidine-producing strain.

General comments and impact:

The authors are correct in pointing out the need for an alternative supply chain for these valuable plant-derived molecules. As the final missing steps in this pathway were only recently reported, the authors' microbial engineering work is timely. The authors also have reason to claim that the manuscript describes a biosynthetic pathway with the most number of heterologous enzymes assembled to date. However, there are numerous concerns (discussed in detail below) with the work that need to be addressed prior to publication.

Data presentation is incomplete or inadequate in some cases. No statistical analyses (e.g. t-tests) are provided, which is concerning given the large inter-replicate variations and error bars seen in some of the graphs. Microscopy image sets are incomplete and inconsistent, and as a result do not provide strong support for some of the authors' conclusions. In some cases, methods used for troubleshooting (e.g., identifying bottlenecks) are unconventional and/or not appropriate – see details in section below.

We thank the reviewer for the thorough and critical review and helpful suggestions. To address all points raised, we have constructed new strains, conducted additional experiments, performed statistical analysis (Student's t-test), and extensively revised the manuscript. Please see our response and revision for each specific point below.

Major questions and points of concern:

1. Choice of starting strain: Brown et al. (2015) previously reported a yeast strain capable of de novo strictosidine production at up to ~500 µg/L. Why was this strain not used directly as the starting point? The authors noted that strictosidine production was initially not detectable in this strain when tested by them, but do not offer any explanation for this – such a discrepancy should be addressed. Was the copy number of heterologous enzymes different from the previously reported strain? Were similar supporting enzymes (CYPADH, SAM2, ZWF1, G8Hco-expressing plasmid) not used, and if not, why not? It is confusing that the authors would not use the same panel of genes and modifications already demonstrated to support de novo strictosidine production as their starting point.

Brown et al. demonstrated the first de novo strictosidine using a yeast strain of BY background. However, we decided to construct using CEN.PK background for the following reasons:

- 1) CEN.PK background strains are a common choice for metabolic engineering given their fast growth compared to other commonly used yeast strains (Canelas et al. 2010, PMID: 21266995) and well-established processes for using these strains in large scale manufacturing (Keasling and Nielsen, 2016, PMID: 26967285).
- 2) High-quality genome and transcriptome sequences are available for these strains (Salazar et al. 2017, PMID: 28961779; Jenjaroenpun et al. 2018, PMID: 29346625)
- 3) Standard synthetic biology tools (promoters, terminators, guide RNA sequences, integration sites, etc) were developed based on this strain background
- 4) Strain 4 from Brown et al. grows extremely slowly due to the knockout of ERG20 (new **Supplementary Fig. 9**) and would not be desirable for further engineering and manufacturing purposes.
- 5) Strain 4 from Brown et al., suffered from instability and did not produce more than 100 µg/L in our initial studies (new **Supplementary Fig. 10**), and was therefore altogether deemed inappropriate for further engineering.

The first strain MIA-BG is not a *de novo* strictosidine strain, but rather relies on externally supplemented geraniol and tryptamine, which might have contributed to the low production of strictosidine since geraniol is both toxic and volatile.

CYPADH (CroADH2) is overexpressed in strains reported in this study.

Initially, only one copy of CroG8H was integrated into the yeast genome, which was sufficient to produce 80 μM of 8-hydroxygeraniol from 100 μM geraniol fed into the growth medium. Also, overexpression of CroG8H from a high copy plasmid only had a modest effect on strictosidine production in strain MIA-BG when fed with geraniol (**Supplementary Fig. 3**).

In the initial strain, overexpression of ZWF1 was not prioritised because it is well documented that glucose-6-phosphate dehydrogenase is primarily controlled by allosteric regulation and inhibited by high NADPH/NADP⁺ (Minard & McAlister-Henn 2005, PMID: 16179340, Llobell et al. 1988, PMID: 3277619). Further, the NADPH requirement for making 500 $\mu\text{g/L}$ of strictosidine is 4-5 μM , equivalent to 0.36-0.54 mg/L glucose metabolised through the oxidative branch of the pentose phosphate pathway (PPP); therefore strictosidine production is unlikely to be largely limited by low NADPH availability. As we have shown in **Fig. 2**, the optimised strain MIA-CH-A2 and strains MIA-CM-3 or MIA-CM-5 could produce much higher strictosidine without overexpression of ZWF1 (this does not necessarily mean that ZWF1 overexpression would not be beneficial for the production of strictosidine at a high concentration such as >100 mg/L).

Similarly, regarding SAM2 overexpression, at the production level of 500 $\mu\text{g/L}$ of strictosidine, the total flux through the methyltransferase (LAMT) was 1 μM . We fed MIA-BG strain with *cis-trans*-nepetalactol and loganic acid, and it could produce 0.39 mg/L and 2.51 mg/L strictosidine, respectively (**Fig. 2b**), indicating the low production in the MIA-BG strain was not mainly due to insufficient SAM pool size (again, this does not necessarily mean that SAM2 overexpression would not be beneficial for the production of strictosidine).

2. Insufficient or inappropriate bottleneck analysis: The authors deduce pathway bottlenecks using fluorescence micrographs and targeted proteomics, which do not provide an accurate picture as they measure limited aspects of protein expression. In microscopy, fluorescence intensity is determined not only by protein expression, but also by folding, localization, microenvironment of GFP, or even small differences in sample preparation, microscope settings, exposure time, cell growth phase, etc. Thus, low or high fluorescence cannot be interpreted as a proxy for expression level. Targeted proteomics is also affected by differences in protein stability during extraction, sample preparation, etc.

In the Results section “Optimization of the de novo strictosidine platform”, the authors compared fluorescence intensity of four P450-GFP fusions and stated that three enzymes show low fluorescence, indicative of low expression. However, there is no standard for comparison – is this low fluorescence in comparison to 7DLH? One could just as easily claim that three of the samples show ‘normal’ fluorescence and 7DLH is abnormally high, depending on the exposure time, settings, etc. Similarly, in the Results section “Functional expression of tabersonine/catharanthine and vindoline modules”, the authors draw conclusions on enzyme expression based on fluorescence intensity in micrographs. Statements about gene expression based on fluorescence data are not accurate.

In the Results section “Optimization of the de novo strictosidine platform”, the authors also argue that their fluorescence observations are consistent with relative protein expression shown by targeted proteomics. However, this is not supported by the data: in Extended Data Fig. 2c, not only is there no standard for comparison to determine what is ‘low’ or ‘high’ expression, but GPPS (which is claimed to be poorly expressed) seems to be the highest-expressed protein, and all four P450 enzymes show comparably average abundances – especially in the absence of any statistical tests. A substrate feeding experiment in the same Results section was also used to deduce bottlenecks, but such experiments assume that fed substrates are transported into cells with equal efficacy, which is rarely the case. Differences in strictosidine titer with different fed precursors may just as easily be explained by different import efficiencies. In each Results section, bottleneck testing should be repeated by overexpressing each enzyme along the pathway and measuring accumulation of intermediates.

We agree with the reviewer that the fluorescence intensity (as judged from the confocal microscope images) should not be used as evidence for poor protein expression. We have removed or revised the statements in the manuscript.

However, the targeted proteomics method was based on the relative quantification of multiple unique peptides within the target protein (Batth et al. 2015, PMID: 25205128). Therefore, we still think that this analysis is informative and indicative of the relative protein abundance, except that this method is unable to distinguish soluble and insoluble proteins.

There was an error in the data presented in the original **Extended Data Fig. 2c**, which has now been corrected in the revised manuscript. In the updated **Extended Data Fig. 2c**, AgrGPPS2 has a relatively low signal, and the truncated CroSTR (tCroSTR) was below the detection limit. Overexpression of tCroSTR from a high copy plasmid significantly improved the strictosidine production, but only when cells were fed with secologanin and tryptamine (**Supplementary Fig. 3**).

We also fully agree with the reviewer that intermediates might be transported into the cell with very different efficiency. This is evident by the observation that feeding loganin, and to a lesser extent loganic acid, only resulted in a very modest improvement in strictosidine production (0.14 mg/L and 0.39 mg/L, respectively). Given that feeding their upstream precursor *cis-trans*-nepetalactol gave much higher production, we concluded that all enzymes downstream of *cis-trans*-nepetalactol were sufficient in converting at least 100 μ M of *cis-trans*-nepetalactol into 2.5 mg/L of strictosidine.

Lastly, to complement the testing of rate-limiting steps based on feeding and protein abundance/localization studies, we have now overexpressed each of the 12x genes involved in the strictosidine module (G8H, 8HGO, ISY, IO, CYPADH, 7DLGT, LAMT, 7DLH, SLS, tSTR, CPR, CYB5) using a 2 μ (high copy) plasmid in the MIA-BG base strain. Here, all strains were cultivated in SC medium lacking uracil (for selection of the 2 μ plasmid); supplemented with 1 mM tryptamine and 0.1 mM of one of the precursors, namely geraniol, 8-hydroxygeraniol, 8-oxogeraniol, *cis-trans*-nepetalactol, loganic acid, loganin and secologanin; and cultivated for 6 days. Loganic acid, loganin, secologanin and strictosidine were analysed in the spent media using LC-MS. Results from the MIA-BG + single gene overexpression and feeding experiment were plotted and included in new **Supplementary Fig. 3**. Overexpression of tCroSTR led to a substantially higher strictosidine production when cells were fed with 100 μ M secologanin, the immediate substrate of this enzyme. However, overexpression of any other single enzyme did not result in significant improvement. We consider it likely that several enzymes show poor functionality, which can undermine the beneficial effects from overexpressing any single upstream enzyme.

Taken together, with the updated extended proteomics data and supplementary figure on single-gene overexpression we hope the reviewer agrees that we can document that strictosidine biosynthesis in yeast can be improved by expression of extra copies of tCroSTR, thus constituting a first rate-limiting step. Meanwhile, we also confirm the note from the reviewer that not all precursors can be expected to be taken up with equal efficiency.

In the Results section “Functional expression of tabersonine/catharanthine and vindoline modules”, the authors expressed additional copies of two consecutive genes in the vinblastine module and concluded that T3O/T3R are the major metabolic control point, despite the fact that Extended Data 9d shows that almost every pairwise extra copy combination substantially increased vindoline levels – implying no standout bottlenecks. The authors then show that the CS/TS branchpoint is a principal limitation, as CS activity appears to far exceed TS activity, but make no efforts to optimize this bottleneck beyond separating the two branches in different strains. The authors should investigate whether tuning the flux split at this CS/TS branch point might enable both branches to cooperate in the same strain.

We agree that the current data from the pairwise overexpression of extra-copy-combinations did not suggest any standout bottlenecks in the vindoline module, and that the metabolic control seems evenly distributed across the whole pathway module from tabersonine to vindoline.

To make a strain with a tunable CS/TS flux split that can produce both vindoline and catharanthine *de novo*, we integrated the *C. roseus* catharanthine synthase (*CroCS*) under the galactose inducible P_{GAL1} promoter, together with other potential bottleneck genes (*AgrGPPS2*, *ERG20-tCroGES*, *CroCPR*, *CroGS*, *CroGO*, *CroPAS*, *CroDPAS*, *CroSAT*, *CroTS*, *CroT16H*, *CroT3O*, *CroT3R*, *CroD4H* and *CroDAT*), into the *de novo* vindoline strain MIA-CW-1, resulting in a new series of strains (MIA-EM-1~3). The P_{GAL1} promoter was chosen so the expression of *CroCS* can be tightly controlled through the addition of galactose (when glucose is depleted) and to avoid competing for the same precursor to *CroTS*. All new MIA-EM strains could produce vindoline and catharanthine *de novo*. Among them, MIA-EM-2 gave the highest production of ~5 µg/L vindoline and ~70 µg/L catharanthine in small-scale (1 mL) fed-batch cultivations; therefore, this new strain was chosen for larger-scale fermentation followed by purification of these two products for chemical coupling to synthesise vinblastine.

3. Lack of robust statistical analysis: Throughout the manuscript, the authors make comparisons between samples to identify optimal orthologs, variants, strains, etc. Yet, no statistical tests are provided to support their comparisons. This is concerning because in many cases, comparisons are made between samples with similar average values and with substantial inter-replicate variation (large error bars relative to mean); see for example Fig. 2c, Extended Data Figs. 2c, 4a, 5, 6, 7, 8c, 9d, 10, 11b. Statistical tests should be performed and shown for all bar graphs to support the authors' comparisons.

We performed statistical analysis (Student's t-tests) and included *p*-values in the source data files. Some major significant changes are also indicated in the updated figures.

In addition, certain graphs have error bars that are unusually large: see for example Extended Data Figs. 5, 6, 7. Can the authors offer an explanation for why such a large inter-replicate variation was observed? In the case of Extended Data Fig. 6, it appears that one replicate for every sample had a reading of zero. This implies some experimental failure or systematic error – is there a reason why one replicate for each sample had zero production? If this was an error in experimental setup, then it should be repeated; otherwise, it is difficult to draw meaningful comparisons between samples with such large error bars.

We have repeated the experiment for **Extended Data Fig. 6** and collected data of higher quality. It was very clear that Vmi8HGO-A was the most active variant under the screening condition, and therefore was chosen to be integrated into the strictosidine strain MIA-CH-A2 (parent strain for the *de novo* strain MIA-CM-3 and MIA-CM-5).

4. Incomplete or ambiguous microscopy: The presentation and/or interpretation of fluorescence micrographs is inadequate to support many of the conclusions. In some cases, images were not taken at sufficient magnification and/or are simply too small to evaluate sub-cellular features (e.g. Fig. 3c, Extended Data Fig. 8d)—these images should be magnified, and/or re-acquired at a higher

(and consistent) magnification. In addition, images should be shown separately as bright field (BF), GFP/fluorophores, and merged in order to clearly discern overlap of features; showing merged images alone (e.g. Fig. 3c) is not sufficient.

In other cases, claims about sub-cellular enzyme localization are not evident using GFP fusions alone. Based on Extended Data Fig. 2b, the authors claim that G8H, IO, and SLS are in the ER membrane but 7DLH is not. In fact, all four P450s appear to be in the ER, although the image for 7DLH appears overexposed, which distorts some of the features. To claim ER localization, the authors should co-image the GFP fusions with an ER marker or stain and ensure that all images are similarly exposed. Based on Fig. 3c, the authors conclude that RseSGD is found in the nucleus while CroSGD is cytosolic. However, CroSGD does not resemble cytosolic distribution and shows concentrated accumulations, which could be the vacuole, perinuclear ER, or inclusions. Similarly, RseSGD only shows central puncta, which are not necessarily nuclear. To validate these claims, the authors should co-image the SGDs with vacuolar and nuclear/DNA stains, and show BF, GFP, and merged images separately.

Moreover, the authors state that the nuclear localization signal (NLS) in SGD is C-terminal, yet they image C-terminal GFP-tagged SGD. C-terminal fusion is likely to mask and interfere with the NLS, precluding accurate determination of native localization. The authors should also image N-terminal GFP fusions of Rse and CroSGD to accurately determine differences in their localization in yeast.

We constructed new strains with a red fluorescent marker localised in either the nucleus, mitochondrion, vacuole or on the ER membrane, and a yEGFP fused to the N-terminus of RseSGD, CroSGD and hybrid variants C₁C₂R₃R₄, C₁C₂R₃C₄, C₁C₂C₃R₄. All these strains were cultivated in YPD medium overnight and washed once before confocal microscopy analysis. New confocal microscope images with higher quality were taken with consistent procedures and identical settings (magnification, voltage, exposure, etc) for all strains (**Supplementary Fig. 5**), which indicate that RseSGD as well as hybrid variants that had the 3rd domain from *R. serpentina* (C₁C₂R₃R₄ and C₁C₂R₃C₄), are forming aggregates/puncta that are localised in the nucleus, while CroSGD and hybrid variants that had the 3rd domain from *C. roseus* (C₁C₂C₃R₄) are localised in the cytoplasm. All imagery (**Supplementary Figs. 1, 4, 5, 7 & 8**) now includes bright-field images and fluorescence images. We didn't manage to fuse yEGFP to the N-terminus of variant R₁R₂C₃R₄ to verify if R₃ is the only domain required for proper SGD function. Yet we want to remind the reviewer that this variant did not produce THA when CroHYS was co-expressed.

5. Incomplete final step: The authors frame the work in the context of a semi-synthetic supply chain for vinblastine, despite the existence of an enzyme (PRX1) to catalyze the final step. Although the authors observed no activity (and no expression, based on proteomics) from the wild-type enzyme in yeast, no attempt was made to rectify its lack of activity. Have PRX1-like enzymes or members of the same class been previously expressed in microbes or in yeast? PRX1 is predicted to be a secreted/extracellular peroxidase – have the authors checked whether it is also being secreted in yeast (and perhaps that is why they observed no cellular protein)? Perhaps removing secretion signals, and/or localizing it to the yeast peroxisome via PTS1, would rectify cellular activity. N- and C-terminal GFP fusions of PRX1 may also shed light on the fate of the protein in yeast. From the presentation it appears that the authors attempted no optimization of this enzyme whatsoever; which is in part concerning with how Figure 1 is presented (see comment #6 below).

Prx1 is reported or predicted to have an N-terminal (N21 and N34) and a C-terminal (C25) vacuole signal peptide. We constructed five different truncated variants of Prx1 (N21Δ, N34Δ, C25Δ, N21ΔC25Δ and N34ΔC25Δ), but none of the truncated variants could convert the catharanthine and vindoline supplemented in the medium to vinblastine (data not shown). We also fused a yEGFP to the C-terminus of the full length and all truncated variants to investigate their subcellular localization. Removal of N21 signal peptide (tCroPRX1_{N21Δ} and tCroPRX1_{N21Δ C25Δ}) also changed the localization of the protein from the cytoplasm to mitochondria (**Supplementary Fig. 8**). Plant peroxidases can couple catharanthine and vindoline into anhydrovinblastine (Goodbody et al. 1988). Unfortunately, our attempt of using the horseradish peroxidase (purchased from Sigma-Aldrich) to catalyse the coupling of catharanthine and vindoline into anhydrovinblastine in the presence of H₂O₂ was unsuccessful (data not shown). On the other hand, the observation that UV-A photo-chemical and Fe(III)-based chemical coupling methods could produce relatively high levels of vinblastine and even higher anhydrovinblastine peaks (although unable to quantify due to lack of an authentic analytical standard) suggested that the chemical coupling method is more favourable for the production of vinblastine.

6. Misleading description and figure: The authors' description of their vinblastine semi-synthesis is misleading; in the final sentence of the Results, the authors claim to demonstrate that the vindoline and catharanthine produced by the yeast can be readily coupled enzymatically or photochemically – but this is not fully demonstrated. First, the authors do not show any semi-synthetic coupling with yeast-produced substrates, only using pure standards instead at higher concentrations. Second, the authors' attempted enzymatic coupling with HRP did not work; only photochemical coupling of the pure standards was successful. It would be more appropriate to state only that photochemical coupling of vindoline and catharanthine standards is feasible in vitro, suggesting that it may be adapted for use with strains producing much higher titers of these substrates in the future.

Similarly, the title of the paper is misleading in that it claims to present a microbial supply chain for vinblastine, when in fact no vinblastine was able to be produced bio- or semi-synthetically. Its two immediate precursors vindoline and catharanthine were produced de novo in separate yeast strains, and at low titers (<1 μg/L) even by the standards of a proof-of-concept study. The title should be changed to more accurately reflect what was shown in this paper (e.g. yeast biosynthesis of vindoline and catharanthine). Figure 1 is similarly misleading in that it shows the last coupling step (between vindoline and catharanthine) happening enzymatically, which gives the impression that this step was reconstructed in yeast in the context of the full pathway. In fact, this step was not shown to be functional in their yeast strains and the synthesis of vindoline and catharanthine occurred in different yeast strains. Figure 1 should be modified to accurately reflect the pathway engineering described in this paper.

These are valid points. We have now revised the figures as the following:

1. In **Fig. 1**, besides Prx1, we added chemical coupling for the synthesis of anhydrovinblastine and vinblastine
2. in **Extended Data Fig. 10**, removed peroxidase and H₂O₂ and only show photo-chemical (UV-A) and Fe(III) chemical coupling methods.

3. We have also revised the results section to clarify that the enzymatic coupling reaction didn't produce detectable vinblastine in our experiment.
4. The best strain MIA-EM-2 can enable one-pot production of ~70 µg/L catharanthine and ~5 µg/L vindoline in small-scale (1 mL) fed-batch cultivations and ~90 µg/L catharanthine and ~13 µg/L vindoline in ambr250 bioreactors. The new results are now included in new **Extended Data Fig. 9**.
5. Lastly, and of utmost importance, we now demonstrate whole-cell biosynthesis of vindoline and catharanthine produced in a single strain, and subsequent Fe(III)-based chemical coupling of these precursors into anhydrovinblastine and vinblastine (**Extended Data Fig. 10b**). Based on these new and important results we have now decided to keep the original title.

Minor points:

1. In the final sentence of the first Results section ("The complete biosynthetic pathway for anti-cancer drug vinblastine"), the authors state that each metabolic module also included *C. roseus* P450 reductase (CPR) and cytochrome b5 (CYB5). Is there evidence that this particular pairing is optimal for the CYPs in this engineered pathway? Other plant CPRs, such as Arabidopsis ATR1, are commonly used in yeast to support diverse plant CYPs (with or without CYB5), and different CPR/CYB5 pairings may exhibit better activity with different CYPs – which is relevant given the large number of CYPs in this pathway. The authors should include a suitable reference (or data) justifying their choice of this redox pairing, and why multiple copies were included; typically one CPR/CYB5 can support numerous CYPs, and CPR/CYB5 overexpression may be disruptive.

All seven P450 enzymes (G8H, IO, 7DLH, SLS, GO, T16H, T3O) are from the native producer *C. roseus*, which is the only known natural producer of the anti-cancer agents vinblastine and vincristine. Therefore, by default we chose to use the CPR and CYB5 from the same species, which from an evolutionary perspective should have no issue interacting and supporting the seven P450 enzymes in this pathway. Further, CroCPR and CroCYB5 were also used in the *de novo* strictosidine production reported by Brown et al. 2015 and worked sufficiently well, as well as in our study. We speculate that the low titers of vindoline and catharanthine are mainly due to 1) poor expression of P450 enzymes and CPR partner, and 2) suboptimal/imbalanced expression of enzymes in this extremely long pathway. However, further optimization will require extensive engineering at another level, which is beyond the scope of this study.

That said, we totally agree with the reviewer that specific P450-CPR pairs might have better activities. However, that optimization requires a huge effort because there are many CPRs from other plants that can be screened. Furthermore, multiple CPRs should be included in the combinatorial library to test the overall effect on the whole pathway, rather than testing each of the 7x P450-CPR pairs individually, because they may also interact and become suboptimal. However, the space is rather large and far beyond the scope of this study, which is to demonstrate the feasibility of expressing such a long biosynthetic pathway in a microbial system, such as yeast.

2. In the third paragraph of the Result section entitled “Optimization of the de novo strictosidine platform”, it seems like the authors omitted one of the titers for strictosidine produced by MIA-BG when feeding nepetalactol. Similarly, the highest strictosidine production in MIA-BG seems to have been miscopied as 1.93 mg/L, but the referenced figure (Fig. 2b) shows it to be 6.15 mg/L. This should be corrected or addressed.

We have corrected the mistakes (Page 5 Line 13-14).

3. It is interesting that strictosidine synthase is natively a vacuolar enzyme in plants, and full-length STRs retaining the vacuole-targeting sequence are more active in yeast. Combined with the observed formation of inclusion bodies with the truncated version, this suggests that vacuolar trafficking may be important (but not essential) for STR folding or maturation. Can the authors provide a hypothesis for the structural basis of these observations? Is STR expected to receive post-translational modifications, such as disulfide formation or N-glycosylation, along its trafficking pathway they may be important for activity? From the micrographs in Extended Data Fig. 4b, even the full-length CroSTR does not exhibit complete localization; some STR appears to be in the vacuole, while another fraction appears to reside near the plasma membrane (perhaps in peripheral ER). The authors may wish to discuss fixing this mis-targeting as a strategy for optimization.

This is very constructive feedback. New images in **Supplementary Fig. 6** show that truncated CroSTR localises to the nucleus in fresh cultures, and only shifts to the cytoplasm following overnight growth. For full-length CroSTR, the localization is less clear. We observe that CroSTR localises sometimes to the ER membrane and sometimes to the vacuole. We have revised the manuscript to report these important spatio-temporal patterns (page 5, line 25-28).

4. In the final paragraph of the Results section “Optimization of the de novo strictosidine platform”, the authors mention several modifications to host geraniol biosynthesis used to increase the pool of this starting substrate. The authors reference several studies which presumably identified these modifications but provide little explanation for their inclusion in this study. For example, why were certain point mutants of ERG20 and GgaFPS included? A brief comment on the rationale for these modifications would improve the clarity of this section.

The yeast native farnesyl pyrophosphate synthetase (encoded by ERG20) catalyses the formation of farnesyl pyrophosphate (FPP), while the double mutant F96W-N127W variant was reported to have abolished FPPS activity and substantially improved monoterpene production in engineered yeast (Ignea et al. 2014, Zhao et al. 2017).

AgrGPPS2 from the grand fir (*Abies grandis*) is a specific geranyl diphosphate synthase (GPPS) (Burke & Croteau 2002) that does not produce farnesyl pyrophosphate. The GgaFPS N144W variant (GgaFPS^{N144W}) had discriminated binding towards GPP as the substrate for the second reaction catalyzed by the enzyme (Stanley Fernandez *et al.* 2000). We combined all these strategies, together with the knock-down of native ERG20, to ensure a strong conversion of DMAPP and IPP into GPP, while maintaining a relatively low production of FPP that is essential as it's required for ergosterol biosynthesis. In addition to explaining this in our manuscript (page 6, line 21-28), we hope that this response further clarifies our engineering strategy.

We have added this technical note below Supplementary Table 1 for heterologous genes (although strictly ERG20 variant is not a heterologous gene).

5. In the Results section “Functionalization and engineering of the gateway SGD”, the authors contrast a previous study in which functional expression of SGD was achieved in yeast with the lack of observed SGD activity in this work, but provide little explanation. Was this previous study able to observe SGD activity in yeast, and if so/not why was functional expression only ‘suggested’? Near the end of this paragraph, the authors suggest that the replacement of the R3 and R4 domains enable higher SGD activity due to complex formation or correct localization – was this experimentally verified? I.e., does the best chimeric SGD localize differently to wild-type CroSGD and RseSGD?

Luijendijk et al. (1996) developed a colorimetric assay and HPLC method for the detection and quantification of cathenamine and its 19-epimer, both derived from deglycosylation of strictosidine by SGD. Since cathenamine and other isoforms of strictosidine are insoluble in water, they form yellowish crystals around the nuclear peripheral in the cell. Geerlings et al (2000) used this characteristic to infer strictosidine aglycone accumulation, and therefore confirmed the SGD activity, when it was expressed in *S. cerevisiae*. We expressed the same SGD gene in yeast and it didn’t have any SGD activity, as evidenced by no strictosidine aglycone production in a yeast strain expressing CroSTR+CroSGD. We only observed production in strains with RseSGD (**Fig. 3**) and together with CroTHAS. When co-expressing CroTHAS, yeast could produce THA only with RseSGD, but not with CroSGD (**Fig. 3a**). Furthermore, and as mentioned above, both SGDs were expressed in yeast but had distinct localization with RseSGD localised in the nucleus and CroSGD localised in the cytoplasm (**Supplementary Fig. 5**). It was unclear to us why CroSGD failed to localise to the nucleus, but Domain 3 seems to have a role because the hybrid variants C₁C₂R₃C₄ & C₁C₂R₃R₄ were localised in the nucleus and C₁C₂C₃R₄ was localised in the cytoplasm. We didn’t manage to fuse the yEGFP to the hybrid variant R₁R₂C₃R₄, which did not have SGD activity when tested for THA production (**Fig. 3c**).

6. Tables of peaks and retention times should be avoided as panels in figures, and shown instead as representative chromatogram traces or heatmaps to facilitate visual comparison (applies to Fig. 3b, Extended Data Fig. 9c). For Fig. 3c, the specific constructs expressed for fluorescence microscopy (i.e. C-terminal GFP fusions) should be indicated in the caption.

We have removed the tables as suggested. Also, strain names that expressed GFP fusion proteins are indicated in the figure caption (**Supplementary Figs. 1, 4, 5, 7 & 8**).

7. In general, the authors may wish to use a numerical yeast strain numbering system, as the present system can be confusing to keep track of lineages, and in some cases are mis-referenced or mis-typed by the authors. For example, Extended Data Fig. 8c includes strain MIA-DA, but this strain does not appear in the strain table (Supp. Table 4). Similarly, in Extended Data Fig. 11b the graph references strains MIA-MF-1 and MIA-MF-2, but these do not appear in Supp. Table 4 (only MIA-MI-1 and -2). Are these omissions or typos?

We apologise for the typos.

We have now constructed new haploid strains (MIA-EMs) to enable one-pot production of catharanthine and vindoline (New Fig. 4). Therefore, we deleted the whole section describing the mated strains for *de novo* catharanthine or vindoline production, and updated the strain table.

8. In the Results section “Functional expression of tabersonine/catharanthine and vindoline modules”, the authors observed leurosine formation in both wild-type and MIA-DE strains fed with catharanthine and vindoline. Was leurosine also detected in media without yeast? If not, are there particular biochemical conditions, cofactors (NAD(P)), etc. present in or produced by yeast cells that support this non-enzymatic step? Would the authors be able to detect this condensation product if yeast media or supernatant containing *de novo* biosynthesized catharanthine and vindoline were assayed? If so, could this approach be adapted for non-enzymatic synthesis of vinblastine?

Leurosine was not detected in blank media without yeast cells. One possibility is that yeast can provide certain cofactors, such as FMN and NAD(P)H that could assist in coupling catharanthine and vindoline to form 3,4-anhydrovinblastine, and further reduction to leurosine or vinblastine. We could not detect any coupling products in the supernatant of strain MIA-EM-2 presumably due to the low catharanthine and vindoline concentrations.

9. In general, many labels and text on graphs are either very crowded, or too small to be legible. In the first case, consider showing x-axis labels in tabular format (as shown in Fig. 2b) – this would benefit Extended Data Figs. 2c, 4a, 9d. In the second case (see e.g. Extended Data Figs. 7b, 12bc), consider expanding the graphs, or omitting extraneous text (many Extended Data chromatogram traces look like screenshots from acquisition software – consider re-processing into simplified trace arrays for clarity).

We thank the review for the good suggestion for better data visualisation. We have regenerated chromatograms of higher quality and ensured readable font sizes throughout (Extended Data Fig. 10b, Supplementary Fig. 11-13).

Referee #2 (Remarks to the Author):

Zhang et al. describe the heterologous production of the valuable monoterpene indole alkaloid (MIA) vinblastine. MIAs possess medically valuable bioactivity, but it is difficult to obtain enough material for clinical usage due to challenges associated with both extraction from the native producing plant and its complicated total synthesis. Thus, researchers have used engineered yeast *Saccharomyces cerevisiae* to produce these MIAs. The authors first optimized a strictosidine-producing yeast strain by eliminating shunt product pathways and testing different promoters and homologues of involved biosynthetic enzymes. Following strictosidine production optimization, the authors engineered the next enzyme involved in the biosynthetic pathway, strictosidine- β -D-glucosidase (SGD), by creating hybrid SGDs composed of different domain combinations. The authors also engineered a catharanthine and vindoline-producing yeast strain through the integration of the plant biosynthetic genes.

The strictosidine and catharanthine/vindoline-producing strains were mated, producing a final strain which produced catharanthine upon glucose feeding. The authors also engineered a separate strain through gene integration to produce vindoline upon glucose feeding. Vindoline and catharanthine were subsequently coupled in a semi-synthetic reaction to produce vinblastine.

The work highlights the extensive toolbox available in metabolic engineering and represents a commendable accomplishment. The engineering performed in this study is impressive. Some issues to address prior to publication:

We thank the reviewer for the positive remarks on our study and the helpful suggestions. Please see our response and revision for each specific point below.

The complete biosynthetic pathway for anti-cancer drug vinblastine

Fig. 1 - The caption indicates that dynamically knocked down genes are marked with red arrows. In the text (Optimization of the de novo strictosidine platform section), the authors indicate down-regulated ERG9 and ERG20, which should be noted on the figure, among other knocked down genes.

We have updated the figure as suggested.

Fig. 1 – The color coding of the genes (orange, green, purple) should be indicated in the caption.

We have updated the figure as suggested.

Supplementary Table 1 – The table should be re-ordered. It appears that the table was arranged to follow the order of the biosynthetic enzymes found in Fig. 1. However, there are discrepancies in this arrangement, such as CYB5 appearing higher on the table than GES, even though CYB5 appears earlier in the biosynthetic pathway, or ADH2 appearing after TDC/STR instead of IO.

We adjusted the order of genes in **Supplementary Table 1** to reflect their order in the biosynthetic pathway.

Optimization of the de novo strictosidine platform

Text - The text states that OYE2 generates a shunt product from 8-hydroxygeraniol, but Fig. 1 implies that OYE2 siphons geraniol.

We have updated the text to be consistent with **Fig. 1**.

Text – In paragraph 3 of the section, the text is missing a value for strictosidine production upon feeding cis-trans-nepetalactol.

We have updated the text with values.

Fig. 2 – In paragraph 3 of the section, the author states that MIA-BG produced 377 ug/L strictosidine, or 0.377 mg/L, while Fig. 2b shows a production of 0.04 mg/L upon feeding 8-hydroxygeraniol. (An aside: are three significant figures warranted? This concern is also applicable to other results in the report.)

We have corrected the number (Page 5, line 10-12) "... when feeding 8-hydroxygeraniol, MIA-BG produced 36.7 µg/L (69.1 nM) strictosidine, while feeding 8-oxogeranial or cis-trans-nepetalactol resulted in 124 µg/L (233 nM) or 2.51 mg/L (4.73 µM) strictosidine, respectively..."

Fig. 2B – Strictosidine production from MIA-BG upon secologanin feeding is 6.15 mg/L, while the text states that the highest obtained strictosidine production in MIA-BG was 1.93 mg/L.

We apologise for the discrepancy and corrected the value (Page 5, line 13-14) : "...the highest strictosidine production in MIA-BG (6.15 mg/L or 11.6 µM) was achieved by feeding secologanin..."

Text – There is varying precision in the reported titers throughout the section. There should be consistency in the precision, preferably to match the precision found in Fig. 2B.

We have corrected the values to be consistent.

Text – In paragraph 4, the author states that the improvement in iridoid titers was likely due dosage rather than synergy between the 8HGOs. In Extended Data Fig. 6, looking at Vmi8HGO-A with the empty vector compared to the data with the addition 8HGO, it looks as if production is slightly higher in the case with the empty vector on average, even if there is a single data point for the HGO combinations that is higher production than the Vmi8HGO-A + EVH case. This sentence should be rephrased to represent this.

We have repeated the experiment and collected new data of higher quality. In the new results, production from expressing Vmi8HGO-A was significantly higher than other single variants and also slightly higher than in combination with Cro8HGO-B or Ptr8HGO-B. We have updated the text (Page 6, line 14-18) to indicate this difference, which was presumably due to the fact that 8HGO can also catalyse the reaction in the reverse direction (converting 8-oxogeranial back to 8-hydroxygeraniol).

Text – In the final paragraph of the section, MIA-CL is referenced in parentheses when it should be referenced directly.

We have revised the sentence "Lastly, to produce strictosidine *de novo* from glucose and tryptophan, we increased geraniol availability by dynamically down-regulating *ERG9* and *ERG20* (both encoding GPP consuming enzymes) using *HXT1* and *HXT3* promoters, respectively (Scalcinati et al. 2012), resulting in strain MIA-CL (**Supplementary Table 2**)."

Text – How do the metabolic engineering efforts performed by the authors compare to efforts performed by other researchers? For example, have other researchers engineered high titers of strictosidine production from secologanin by yeast, and if so, how do the values compare? If there is no direct comparison for strictosidine production between this manuscript and other reports, then a feeding experiment from previous strictosidine production reports should be replicated to demonstrate the high titers obtained from this strain. If performed, the experimental results should be integrated into Fig. 2.

Apart from a recent heteroyohimbine study in yeast (Liu et al., 2022, PMID: 35060115), there is only one previous study on strictosidine production in yeast (Brown et al. 2015). We tested the same strain (Strain 4) with and without the overexpression of CroG8H from a 2 μ (high copy) plasmid and compared it to strain MIA-CM-5. Under the same growth condition, Strain 4 produced 89 μ g/L and 98 μ g/L of strictosidine in SC and SC-URA, respectively, which our strain MIA-CM-5 produced 2.1 mg/L and 1.5 mg/L of strictosidine in SC and SC-URA, respectively (**Supplementary Fig. 10**), which are 23.9- and 15.3-fold higher than Strain 4 (Brown et al. 2015).

Extended Data Fig. 3 – Recommend deleting some examples of gene knock out, as that information can be found on Fig. 1. The additional space can be used to provide an example of a promoter swap, for example, or to draw out more explicitly the genetic knock out strategy.

We have modified **Extended Data Fig. 3** as recommended. We removed the deletion of ATF1 OYE2, OYE3, ADH6 and ARI1, and added subpanels to show genetic strategies for swapping truncated with full-length CroSTR, as well as replacing P_{ERG20} with P_{HXT3}.

Functionalization and engineering of the gateway SGD

Extended Data Fig. 7C – The text states that 15 homologues yielded THA production greater than 0.1 mg/L and that 42 CroSGD homologues were tested. These reported numbers do not match what is observed in the figure.

We have corrected the numbers and revised the text (Page 7, line 20-22) as “... when testing 46 sequence homologues of *CroSGD*, *RseSGD* yielded 4.02 mg/L (11.4 μ M) THA from feeding 100 μ M secologanin and 1 mM tryptamine, and another 15 homologues yielded >0.100 μ M or 35.2 μ g/L (0.1% yield from 100 μ M secologanin feeding) THA (**Extended Data Fig. 7b**).”

Fig. 3 – Fig. 3A and Fig. 3B conveys similar information and only one figure is needed. The peak area data does not provide as much information as the bar graph, so the reviewer recommends eliminating Fig. 3B.

We apologise for causing the confusion. In the original version, **Fig. 3b** shows that *RseSGD*, but not *CroSGD* expressed in strain MIA-BJ had accumulated peaks that had exact mass matching with strictosidine aglycone stereoisomers, one of which is the precursor for THA (**Fig. 3a**). To avoid confusion, we have removed panel **3b**.

Fig. 3C – The implied failure of THA production upon *CroSGD* expression is presumably due to localization issues, but the authors don't discuss this. Recommend commenting on the lack of THA production upon *CroSGD* expression.

We have added text (Page 7, line 26-28) speculating that the lack of THA production by *CroSGD* was primarily due to mis-localization of this enzyme to the cytoplasm.

Fig. 3E – Fig. 3E should be cited in the text at the end of the paragraph of this section. Also, is sequence information available for what the authors defined as the four domains? If not, this

information should be provided, potentially in an extended data figure or added to Supplementary Table 1.

We have revised and cited **Fig. 3c** (old **Fig. 3e**) in the text (Page 7, line 32 - Page 8 line 1). Sequences of 4 domains for both *CroSGD* and *RseSGD* are now listed in **Supplementary Table 3**.

Extended Data Fig. 7 -This recommendation applies to the entire manuscript. Higher resolution chromatograms should be used in the images, as they are currently difficult to interpret.

We have regenerated chromatogram figures with higher quality and ensured readable font sizes throughout (**Extended Data Fig. 10b, Supplementary Figs. 11-13**).

Extended Data Fig. 7 – The figure caption refers to Supplementary Table 2, which corresponds to a list of chemical standards used in the experiment. The authors were likely referring to Supplementary table 1.

We have corrected the error.

Functional expression of tabersonine/catharanthine and vindoline modules

Extended Data Fig. 8B – Similar to previous comments, higher quality chromatograms are recommended. Instead of screenshots, the raw data points can be extracted, and the authors can generate these plots.

We have regenerated chromatograms of higher quality, shown in **Supplementary Fig. 11-13**.

Extended Data Fig. 9C – A chart showing peak areas is not the best representation of the data. Suggest presenting a chromatogram with labelled peaks or a bar graph showing relative quantities.

We have removed the peak area and replaced it with a heat map (new **Supplementary Fig. 6e**).

De novo synthesis of catharanthine, tabersonine and vindoline

Text – How was the mating process performed? How did the authors confirm successful mating?

Two haploid strains, each carrying a plasmid with a different auxotrophic marker, were mixed and streaked on an agar plate with synthetic drop-out medium lacking corresponding two amino acids to select the mated diploid strain that carry both plasmids.

However, we decided to delete the whole section describing the mated strains for *de novo* catharanthine or vindoline production, because we have now constructed new haploid strains (MIA-EMs) to enable one-pot production of catharanthine and vindoline (New **Fig. 4**).

Supplementary Table 4 – It can be difficult to keep track of the usage for all of the different yeast strains. Can an additional column be added to the table to annotate the desired end product or something to indicate usage? In its current state, it can be difficult for readers to determine the differences between yeast strains from just looking at the gene names.

This is a valid point. We have modified the strain table (new **Supplementary Table 2**) by adding a column indicating the final MIA product(s) of the strain.

Extended Data Fig. 11B – The bar graphs read MIA-MF-1 and MIA-MF-2 when they should be MIA-MI-1 and MIA-MI-2.

Thanks for spotting the typos.

We have now constructed new haploid strains (MIA-EMs) to enable one-pot production of catharanthine and vindoline (New **Fig. 4**). Therefore, we deleted the whole section describing the mated strains for *de novo* catharanthine or vindoline production, and updated the strain table.

Text – What were the yields associated with the reactions to produce vinblastine directly from catharanthine and vindoline (for both the chemical coupling and enzymatic reactions)?

The enzymatic coupling reaction did not yield any detectable vinblastine. We have calculated the vinblastine yield on pure vindoline and catharanthine standards are around 4% and 0.4%, respectively. The yields for chemical coupling using catharanthine and vindoline purified from fermentation broths were substantially lower. While we were unable to precisely quantify due to complex matrix effects, we estimate the yields to be 0.25% from vindoline and 0.035% from catharanthine, respectively.

Fig. 4 – Can an additional panel be added to incorporate aspects of Extended Fig. 12 into the figure? The production of vinblastine is core to this manuscript and should be reflected in the main figures.

Previously, the photo-chemical coupling was carried out using catharanthine and vindoline standards, either in Tris buffer or growth medium. We have now grown strain MIA-EM-2 in ambr[®]250 bioreactors (in 4 replicates) and performed the coupling reaction using catharanthine and vindoline produced in the fermentation, which gave the production of 23.9 µg/L vinblastine in the chemical coupling reactions. This result is now included in New **Extended Data Fig. 10b**.

Discussion

Text – Have the authors tried isolating and confirming leurosine formation from yeast? If so, could leurosine be isolated and used to produce vinblastine through the proposed epoxide reduction and Fe(III)-based oxidation procedure? It would be valuable to compare the yields from this process to the yields from the methods shown in Extended Data Fig 12. If these experiments are performed, an additional Extended Data Fig. 13 can be created comparing the yields of the three pathways from catharanthine and vindoline to vinblastine.

We have not purified any by-products, which requires a much larger volume of fermentation broth given the low titers and other expertise in the downstream processing. Since there is an established process to purify vindoline and catharanthine from *Catharanthus roseus* (leaves) and subsequently couple them into vinblastine on an industrial scale, the main focus of this study is to create a microbial platform to produce vindoline and catharanthine by yeast

fermentation, and demonstrate the feasibility to chemically couple these two precursors into the final drug vinblastine.

Extended Data Fig. 10 – The figure shows difficulty expressing PRX1 in yeast. Can the authors comment on the extent of engineering attempts? Is there any homologue that can be utilized instead? Alternatively, have the authors tried engineering PRX1 into different yeast strains besides the MIA-DJ strain? Even if the entire pathway cannot be refactored into a singular yeast strain, it would be extremely powerful if vinblastine could be produced strictly from feeding yeast without reliance on purified enzyme or synthetic methods. While this manuscript is already an impressive feat of metabolic engineering, de novo production of vinblastine would complete the story.

Please see below the response to Reviewer #1 main point 5.

Prx1 is reported or predicted to have an N-terminal (N21 and N34) and a C-terminal (C25) vacuole signal peptide. We constructed five different truncated variants of Prx1 (N21Δ, N34Δ, C25Δ, N21ΔC25Δ and N34ΔC25Δ), but none of the truncated variants could convert the catharanthine and vindoline supplemented in the medium to vinblastine (data not shown). We also fused a yEGFP to the C-terminus of the full length and all truncated variants to investigate their subcellular localization. Removal of N21 signal peptide (tCroPRX1_{N21Δ} and tCroPRX1_{N21Δ C25Δ}) also changed the localization of the protein from the cytoplasm to mitochondria (**Supplementary Fig. 8**). Plant peroxidases can couple catharanthine and vindoline into anhydrovinblastine (Goodbody et al. 1988). Unfortunately, our attempt of using the horseradish peroxidase (purchased from Sigma-Aldrich) to catalyse the coupling of catharanthine and vindoline into anhydrovinblastine in the presence of H₂O₂ was unsuccessful (data not shown). On the other hand, the observation that UV-A photo-chemical and Fe(III)-based chemical coupling methods could produce relatively high levels of vinblastine and even higher anhydrovinblastine peaks (although unable to quantify due to lack of an authentic analytical standard) suggested that the chemical coupling method is more favourable for the production of vinblastine.

Text – For microbial production factories to become integrated into the vinblastine production supply chain, they should be compared to current standard practices. How much starting material produces how much vinblastine? What is the time scale of starting materials procurement and vinblastine production with this method compared to standard plant extraction processes? These questions can be addressed by an experiment where the optimized strains are fed glucose and allowed to grow, and the catharanthine and vindoline coupling reaction is performed using whatever material is obtained from the feeding. While this would not be a direct comparison to current industrial standards due to the lack of development of this process, it could strengthen the argument that this platform is actually a viable alternative to current methods.

It is very difficult to compare our proof-of-concept microbial process with the well-established industrial process, in which vinblastine is produced by chemical coupling of vindoline and catharanthine purified from *C. roseus*. We don't have knowledge on the technical details, such as process duration and yield of the chemical coupling reaction, except it is reported that 500 kg of dried *C. roseus* leaves is required to produce 1 gram of vinblastine. In our fed-batch fermentation, in total 78 g/L simple sugars and 7.5 g/L amino acid mixture was used to produce 91.3 ug/L catharanthine and 13.2 ug/L vindoline, which gave roughly a 0.000122%

overall yield. This translates to ~818 kg of starting materials for the production of 1 gram of vindoline and catharanthine (or 1 gram of vinblastine if two precursors can be produced at 1:1 ratio and 100% converted to vinblastine). Undoubtedly, the titer/yield of vinblastine on starting materials (glucose, amino acids, etc) of the microbial process will need to be further optimised to be economically viable. Nevertheless, we envision that an substantially optimised microbial process is an alternative of catharanthine and vindoline production by plant extraction and fully compatible with the current supply chain of vinblastine. It is the experience of one of the co-authors (JDK), who has translated many laboratory technologies to companies, that the titers, rates and yields of such a process would be substantially increased by the company so that the microbial process is orders of magnitude less expensive than extraction and purification from a plant.

Supplementary Tables

Tables should be reviewed and edited for consistent structure and appearance (ex. Line widths, heading format, etc.)

We have edited all supplementary tables to have a consistent appearance.

Referee #3 (Remarks to the Author):

This manuscript by Zhang et al reports the development of a yeast strain to produce vindoline and catharanthine from glyucose and tryptophan. These natural products are the immediate precursors of vinblastine, a critical anticancer drug that faces regular shortages due to its unsustainable and unreliable manufacturing process based on vinca plant extracts. While the authors were not able to synthesize vinblastine in their yeast strain, they demonstrate the ability to produce this drug in vitro from the vindoline and catharanthine produced in yeast. This establishes the viability of a microbial manufacturing process for vinblastine and opens the door to synthesizing potentially thousands of monoterpane indole alkaloids (MIAs) to test for new bioactivities. This important and impressive (their strain contains 44 genetic edits) body of work should be published in a high impact journal, such as Nature, but the authors should clarify the following questions:

1) The abstract states that their strain contains 44 genetic edits, but they say this includes expression of 35 heterologous genes as well as deletions, knock-downs, and overexpression of 10 endogenous genes (a total of 45 gene edits). This obviously does not add up.

We have revised the text to be consistent.

2) Most of their experiments rely on feeding pathway intermediates to yeast strains or measuring their concentrations in fermentation media from strains engineered to produce them. However, there is no discussion on the permeability of the plasma membrane to these metabolites. How well do these transverse the cell membrane, and could it be that many of these intermediate metabolites, when they are trying to produce them, actually accumulate inside the cell? Especially glycosylated metabolites such as secologanin and strictosidine raise questions about their ability to

cross membranes. The authors should provide references or experimental data to answer these questions.

This is a valid point. We have observed poor permeability for loganic acid and loganin (since feeding these two precursors resulted in lower production compared to feeding its upstream precursor *sic-trans-nepetalactol*, **Supplementary Fig. 2**), but there was no evidence indicating a poor uptake/secretion of secologanin and strictosidine. We extracted metabolites from cell pellets using pure methanol, and no significant amount of secologanin or strictosidine was detected (data not shown).

3) In page 6, The authors say that MIA-AU was designed to down-regulate *ERG20* in a strain previously reported to produce strictosidine (Brown et al, 2015), yet they did not observe strictosidine production in MIA-AU (unless supplied with geraniol). Two questions arise: Why did they move from *ERG20* deletion (strain from Brown et al, 2015) to downregulation, if the previous strain already produced strictosidine? How did they downregulate *ERG20* in MIA-AU? Table S4 lists *ERG20* engineered to be controlled by *REV1* promoter, but not much else is mentioned about it. Is this a weaker promoter than *ERG20*? Is it repressed by something?

We tried to delete *ERG20* but several attempts were unsuccessful. The *de novo* strain from Brown et al. 2015 grows extremely slowly (MIA-CL also grows much slower than parent MIA strain) therefore we concluded a knockdown is a better strategy. *P_{REV1}* promoter is a well characterised constitutively weak promoter (Lee et al. 2013, PMID: 24038353). Strain MIA-CL grows slower than its parent strain MIA-CH-A2, as a result of the knockdown of *ERG20* (**Supplementary Fig. 9**)

4) At the bottom of page 6, the authors report poor expression of GPPS, FPS, and GES, but Extended Data Fig 2c actually shows equal or greater peptide signals for these proteins than tHMG2, which is known to express well in yeast. Similarly, the authors say that Extended Data Fig. 10 suggests low expression of TS. However,

There was a mistake in **Extended Data Fig. 2**. The correct figure shows that the expression of GPPS, FPS, and GES are lower than that of tHMG1.

5) In line 11 of page 7 there is missing information. They write “feeding 8-oxogeraniol or *cis-trans-nepetalactol* resulted in 124 µg/L or ___?___mg/L strictosidine,”. How many mg/L?

We have revised the text and added the missing values.

6) The following statement that these results indicate that 8HGO and ISY are major bottlenecks is confusing. How do these results indicate the bottleneck is not upstream? After all, they had to supply these precursors to increase production. These two enzymes are still involved when producing strictosidine from 8-hydroxygeraniol, so it is also unclear how they could reach this conclusion from that comparison. Could it all be differences in the ability of the yeast to import these metabolites (see question 2 above)? Finally, if they clarify how they can conclude from these data that the bottleneck is downstream of the fed intermediates, why do they pinpoint on these two enzymes? Couldn't it be any enzyme downstream from these intermediates all the way to strictosidine (e.g., IO, 7DLGH, LAMT, SLS, STR)? Without measuring the accumulation of all these intermediates, how do they know?

This is a valid point. Please see response to reviewer #1 major point 2.

7) In the last paragraph of page 7, it is not clear what four cyclases they are referring to. MLPs and NEPS2 are not described in any detail. Are they alternatives to ISY? Furthermore Extended Figure 3b, which they reference, does not show any data nor clarify this question.

We have revised the text and stated that NEPS and cyclases are “critical enzymes involved in the cyclisation of 8-oxogeranial to nepetalactol (Lichman et al. 2019, PMID: 30531909; Lichman et al. 2020 PMID: 32426505)”.

8) The experiment in Fig3d,e is puzzling. It is not clear why the authors thought it would be a good idea to shuffle the domains between the two SGDs (CroSGD and RseSGD) or if they gained anything from it (differences in THA production between the RRRR and CCRR don't appear to be statistically significant). A stronger justification for this experiment seems warranted.

It was very striking to observe that despite sharing a high protein identity (70%), RseSGD and CroSGD, when co-expressed with *CroTHAS*, led to such distinct THA production. Therefore, we opted to shuffle the two SGD variants and tried to find the region that is responsible for the mis-localization and lack of SGD activity.

9) The fluorescence microscopy images in Fig3c are not very convincing. CroSGD does not seem to be really homogeneously dispersed in the cytosol, as the authors claim. The RseSGD seems to be subcellularly compartmentalized, but without a nuclear localization marker (e.g., DAPI) it is not clear how the authors can conclude this.

We constructed new strains that also express a red marker target to the nucleus and SGD homologs (Rse, Cro) and their hybrid variants (CCRR, CCCR, CCRC) with GFP fused to the N-terminus. It is very clear that variants that has 3rd domain from Rse (RseSGD, CCRR, CCRC) are localised in the nucleus and gave high production of THA, while variants that has 3rd domain from CroSGD including CCCR were localised in the cytoplasm. RRCR didn't produce THA, however we didn't manage to fuse the yEGFP to the N-terminus of this variant to confirm its cytosolic localization.

10) Did they have to purify the vindoline and catharanthine produced by fermentation to produce vinblastine in vitro, or did they just use the fermentation supernatant? If some purification was involved, then they should include the method.

In the previously submitted manuscript, vindoline and catharanthine standards were used in the photochemical coupling experiment to demonstrate the feasibility. We have now constructed new strains that could produce vindoline and catharanthine in one-pot fermentation at much higher titers. We purified catharanthine and vindoline from the fermentation broths and repeated the chemical coupling, and could detect 23.9 µg/L vinblastine and unknown amount of anhydrovinblastine (due to lack of an authentic standard) produced in the reaction.

11) Was the compound with formula C₄₆H₅₆N₄O₉ produced only when integrating the vinblastine module or also in the parental strain without PRX1. If not, why do they suggest non-enzymatic leurosine formation?

Leurosine was also formed in the wild type strain fed with high concentrations of catharanthine and vindoline (with only Cas9 expressed), therefore likely to be a non-enzymatically coupled product of these two precursors (although we can not exclude the possibility that this coupling reaction involves one or more yeast endogenous enzymes).

12) With so many genetic modifications, the authors should compare the growth rates of their final strains with wild type, so that future researchers know what to expect when reproducing their results.

We have measured the growth rate for all engineered strains and compared them to the reference strain. Results are shown in the new **Supplementary Fig. 9**.

13) It would be valuable to compare the final titers they obtain for vindoline and catharanthine with those obtained from vinca plant extracts.

The same point was raised by reviewer #2

It is very difficult to compare our proof-of-concept microbial process with the well-established industrial process, in which vinblastine is produced by chemical coupling of vindoline and catharanthine purified from *C. roseus*. We don't have knowledge on the technical details, such as process duration and yield of the chemical coupling reaction, except it is reported that 500 kg of dried *C. roseus* leaves is required to produce 1 gram of vinblastine. In our fed-batch fermentation, in total 78 g/L simple sugars and 7.5 g/L amino acid mixture was used to produce 91.3 ug/L catharanthine and 13.2 ug/L vindoline, which gave roughly a 0.000122% overall yield. This translates to ~818 kg of starting materials for the production of 1 gram of vindoline and catharanthine (or 1 gram of vinblastine if two precursors can be produced at 1:1 ratio and 100% converted to vinblastine). Undoubtedly, the titer/yield of vinblastine on starting materials (glucose, amino acids, etc) of the microbial process will need to be further optimised to be economically viable. Nevertheless, we envision that a substantially optimised microbial process is an alternative of catharanthine and vindoline production by plant extraction and fully compatible with the current supply chain of vinblastine. It is the experience of one of the co-authors (JDK), who has translated many laboratory technologies to companies, that the titers, rates and yields of such a process would be substantially increased by the company so that the microbial process is orders of magnitude less expensive than extraction and purification from a plant.

Reviewer Reports on the First Revision:

Referees' comments:

Referee #1 (Remarks to the Author):

The authors have provided substantial additional data in the resubmitted manuscript which, taken together with their responses to individual concerns, address all major points initially raised. The authors provided sufficient justification for their choice of starting strain; performed several additional bottleneck analyses that collectively (along with proteomics and higher-quality microscopy data) provide a more accurate picture of flux limitations in their strains; incorporated a two-phase fermentation system to enable control over the tabersonine-catharanthine bifurcation using an inducible promoter; repeated experiments with high variability and indicated statistical significance on graphs; provided higher-quality microscopy datasets with channel-delineated images and organelle markers, which now provide stronger evidence for their claims of specific enzyme/variant localization; established that preliminary structural engineering of Prx1 is not sufficient for activity in

yeast (indicating that this issue is better suited for a follow-up study); and importantly demonstrated semi-synthetic coupling of yeast-biosynthesized catharanthine and vindoline to vinblastine. The authors have also addressed all original minor points, considering the substantial additional data provided and the scope of the current study.

Although the described work does not report any groundbreaking new discoveries (in the context of the biosynthetic pathway or the enzymes themselves) or technologies/approaches (in the context of engineering these types of complex pathways in microorganisms), as the authors claim the engineered pathway is the longest demonstrated in a heterologous host to date. The work will act as a starting point for yeast engineering efforts towards unlocking the enormous chemical space of MIAs of medicinal relevance.

Please note the following remaining points below that can be addressed:

Remaining minor points:

1. In the Introduction, line 16, there appears to be a missing word after “vincristine and”.

>> We are sorry for the confusion. “.. vinblastine were..” has now been added in the sentence and grammar updated.

2. In the section “The complete biosynthetic pathway for anti-cancer drug vinblastine”, the authors may wish to highlight the complex trajectory of their pathway, with two different starting compounds (and thus two upstream precursor pools to optimize), and a divergence/convergence point downstream with TS vs. CS. These challenges make the engineering involved in establishing a functional pathway more impactful.

>> This is a great comment. We have now included the following statement in the revised manuscript:

“The 31-step vinblastine pathway is remarkably complex including dual inlet precursor supplies from tryptophan and geranyl pyrophosphate (GPP) as well as divergence between catharanthine and vindoline supplies towards vinblastine biosynthesis”

3. For all legends describing fluorescent micrographs, the authors should indicate what construct was used for the organelle markers, instead of stating that the red marker was on a specific organelle (e.g., the red marker was mCherry-HDEL which localizes to the ER).

>> Valid point. In the legends for the relevant Supplementary Figures, we have specified the tags used for mCherry and stated their localizations.

4. On page 28 of the Supplementary Information file, in the text preceding Supplementary Fig. 2, the authors use the term ‘update’ when they may instead mean ‘uptake’.

>> Thanks for spotting this typo. “update” has now been replaced with “uptake”.

5. On page 5 of the main article file, lines 28-30, the authors suggest that substrate localization to the vacuole and/or its acidic pH are responsible for the improved STR activity in the vacuole vs. the cytosol. However, it also bears mentioning that post-translational modifications or specific folding steps encountered along the ER-to-vacuole trafficking pathway may support (but are not required for) STR activity.

>> Valid point. We have now included a sentence disclosing these possibilities:

“Alternatively, post-translational modifications or specific folding steps encountered along the ER-to-vacuole trafficking pathway may support STR activity.”

6. In Extended Data Fig. 6, even the empty vector negative controls show appreciable strictosidine production. Is this explained by promiscuous activity of native yeast oxidoreductases? If so, the authors should include this clarification in the text.

>> Good point. Yeast has multiple ADHs, and we indeed consider it likely that yeast ADHs can catalyze 8-hydroxygeraniol → 8-oxogeraniol conversion. We have included this speculation in the revised manuscript.

“Empty-vector control strains produced 1.86 mg/L (4.38 μM) metabolites, indicating native yeast oxidoreductase activity on 8-hydroxygeraniol”

7. On page 8 of the main article, lines 21-24, the authors claim from Supp. Fig. 6c that the pairing of T3O + T3R yielded the highest increase in vindoline titer – yet the figure suggests that T16H2 + 16OMT yielded a greater improvement. This should be clarified.

>> Good point. We are sorry about the confusion. The best performing OX couple was indeed T16H2 + 16OM. Manuscript has now been updated.

8. Although the authors were unable to demonstrate PRX1 activity in vivo, it would be useful to suggest avenues of future study to rectify this step in the discussion—establishing a complete in vivo pathway to vinblastine in the future would be a significant advantage over semi-synthesis.

>> Valid point to discuss. We would suggest 1) testing different peroxidase homologs, and 2) compartmentalization in peroxisome (higher H₂O₂). We have now revised the manuscript Discussion:

“While it may one day be possible to identify a screening more PRX1 homolog that successfully catalyzes the reaction when compartmentalized in the peroxisome, nevertheless this study provides the first demonstration of a microbial supply chain for a complex essential anti-cancer medicine.”

Referee #2 (Remarks to the Author):

The manuscript has been satisfactorily revised to address my concerns.

>> We thank the reviewer for acknowledging this.

Referee #3 (Remarks to the Author):

The authors addressed all comments raised by this reviewer. This impressive body of work should be accepted for publication.

>> We thank the reviewer for the positive remarks and support.